# Enhanced Wound Healing Potential of *Spirulina platensis* Nanophytosomes: Metabolomic Profiling, Molecular Networking, and Modulation of HMGB-1 in an Excisional Wound Rat Model

**DOI:** 10.3390/md21030149

**Published:** 2023-02-24

**Authors:** Hanan Refai, Amira A. El-Gazar, Ghada M. Ragab, Doaa H. Hassan, Omar S. Ahmed, Rehab A. Hussein, Samah Shabana, Pierre Waffo-Téguo, Josep Valls, Asmaa K. Al-Mokaddem, Heba Mohammed Refat M. Selim, Einas Mohamed Yousef, Sahar K. Ali, Ahmed Salman, Hagar B. Abo-Zalam, Rofida Albash

**Affiliations:** 1Department of Pharmaceutics, College of Pharmaceutical Sciences and Drug Manufacturing, Misr University for Science and Technology, Giza 12585, Egypt; 2Department of Pharmacology and Toxicology, Faculty of Pharmacy, October 6th University, Giza 12585, Egypt; 3Department of Pharmacology and Toxicology, Faculty of Pharmacy, Misr University for Science & Technology (MUST), Giza 12585, Egypt; 4Department of Pharmaceutical Analytical Chemistry, College of Pharmaceutical Sciences and Drug Manufacturing, Misr University for Science and Technology, Giza 12585, Egypt; 5Bordeaux INP—Institut des Sciences de la Vigne et du Vin, University of Bordeaux, UFR des Sciences Pharmaceutiques, Unité OENO, UMR 1366 INRAE, CS 50008—210, Chemin de Leysotte, 33882 Villenaved’Ornon, France; 6Pharmacognosy Department, National Research Centre (NRC), Giza 12622, Egypt; 7Department of Pharmacognosy, College of Pharmaceutical Sciences and Drug Manufacturing, Misr University for Science and Technology, Giza 12585, Egypt; 8Bordeaux Metabolome, MetaboHUB, PHENOME-EMPHASIS, 33140 Villenaved’Ornon, France; 9Pathology Department, Faculty of Veterinary Medicine, Cairo University, Giza 12613, Egypt; 10Department of Pharmaceutical Sciences, Faculty of Pharmacy, Al-Maarefa University, Diriyah, Riyadh 13713, Saudi Arabia; 11Microbiology and Immunology Department, Faculty of Pharmacy (Girls), Al-Azhar University, Cairo 35527, Egypt; 12Histology and Cell Biology Department, Faculty of Medicine, Menoufia University, Shebin Elkom 3251, Egypt; 13Department of Clinical Pharmacology, Faculty of Medicine, Zagazig University, Zagazig 44519, Egypt; 14Department of Anatomy and Histology, Faculty of Medicine, The University of Jordan, Amman 11942, Jordan; 15Department of Anatomy, Faculty of Medicine, Menoufia University, Shebin Elkom 3251, Egypt

**Keywords:** phytosomes, blue green algae, wound healing, molecular docking, HMGB-1, molecular networking

## Abstract

Excisional wounds are considered one of the most common physical injuries. This study aims to test the effect of a nanophytosomal formulation loaded with a dried hydroalcoholic extract of *S. platensis* on promoting excisional wound healing. The *Spirulina platensis* nanophytosomal formulation (SPNP) containing 100 mg PC and 50 mg CH exhibited optimum physicochemical characteristics regarding particle size (598.40 ± 9.68 nm), zeta potential (−19.8 ± 0.49 mV), entrapment efficiency (62.76 ± 1.75%), and Q6h (74.00 ± 1.90%). It was selected to prepare an HPMC gel (SPNP-gel). Through metabolomic profiling of the algal extract, thirteen compounds were identified. Molecular docking of the identified compounds on the active site of the HMGB-1 protein revealed that 12,13-DiHome had the highest docking score of −7.130 kcal/mol. SPNP-gel showed higher wound closure potential and enhanced histopathological alterations as compared to standard (MEBO^®^ ointment) and *S. platensis* gel in wounded Sprague-Dawley rats. Collectively, NPS promoted the wound healing process by enhancing the autophagy process (LC3B/Beclin-1) and the NRF-2/HO-1antioxidant pathway and halting the inflammatory (TNF-, NF-κB, TlR-4 and VEGF), apoptotic processes (AIF, Caspase-3), and the downregulation of HGMB-1 protein expression. The present study’s findings suggest that the topical application of SPNP-gel possesses a potential therapeutic effect in excisional wound healing, chiefly by downregulating HGMB-1 protein expression.

## 1. Introduction

Wound injuries heal through the regeneration and replacement processes of damaged connective tissue at the wound bed, which are the result of dynamic and intricate interactions involving cellular, molecular, biochemical, and physiological functions [1]. Normally, wound healing occurs through four overlapping phases: hemostasis, inflammation, proliferation, and remodeling [2]. At the moment of injury, blood vessel constriction occurs during the hemostasis phase, and a blood clot is formed to prevent further blood loss [3]. This is followed by the inflammation phase, where the blood vessels’ permeability increases, thus permitting the enzymes and immune cells to reach the wound. Moreover, various growth factors and cytokines are released to stimulate fibroblasts, keratinocytes, and endothelial cells to restore the damaged blood vessels [2]. Then comes the proliferative phase, where keratinocytes and fibroblasts move into the wound bed to fill the wound gap, together with re-epithelialization at the wound edge and reformation of the skin barrier. In addition, angiogenesis occurs, and new blood vessels are formed at the wound bed. The final stage involves tissue remodeling and maturation, which causes the skin to be fully recovered [4].

Providing an efficient agent for wound dressing is always an urgent concern in modern medicine. Herbal medicines and phytoconstituents have been demonstrated to be useful in the treatment of a wide range of health conditions, including wound healing. Medicinal plants, such as Curcuma longa L., Terminalia arjuna Roxb., Centella asiatica L., Bidens pilosa L., Aloe barbadensis Mill., and Rauwolfia serpentine L., have confirmed wound healing activity and are found to be effective in the treatment of wounds [5].

*S. platensis* is a microalgae belonging to the Oscillatoriales. It blooms naturally in alkaline lakes. It is an important source of protein as it contains about 70% of easily digestible proteins, [6], and is rich in χ-linolenic acid, vitamins, polyphenols, phycocyanin, and various minerals [7]. Numerous research have shown that *S. platensis* extracts have hypolipidemic, antimutagenic, antifungal, antibacterial, antiviral, anticancer, antiallergic, immune-enhancing, and hepatoprotective properties [8,9,10]. Furthermore, its therapeutic implications in cases of diabetes, arthritis, anemia, cardiovascular diseases, and cancer were reported [11]. When reviewing the literature, a few studies were found concerning the use of *S. platensis* extract in skin wound healing [12]. According to Elbialy et al. (2021), cultivated *S. platensis* is effective in enhancing wound angiogenesis and collagen deposition. It decreases histopathological morphologic change, prevents scarring by up-regulating angiogenic bFGF and the vascular endothelial growth factor (VEGF) genes, and down-regulates the genes TGF-β and α-SMA that are related to scar formation [13]. In addition, a study conducted by Jung et al. (2016) reported the efficiency of *S. platensis* in enhancing wound regeneration by increasing fibroblast viability and enhancing the antioxidant mechanism of fibroblasts under oxidative stress [14]. Many studies have been conducted to investigate the effects of *S. platensis* aqueous extract on wound healing [12,15,16], but only a few have discussed in detail the efficiency of *S. platensis* alcoholic extract.

A huge scientific body discussed the efficacy of herbal medicines and their various extracts in alleviating illnesses such as inflammation, diabetes, cancer, and others through an array of biologically potent phytoconstituents [17]. However, there are some limitations to the efficient delivery of these compounds to the target tissues. The heavy molecular structure of these compounds and their poor lipid solubility restrict their ability to pass through biological membranes, which are rich in lipids [18]. Therefore, a large standard dose is usually required, which restricts the widespread use of phytomedicines in the pharmaceutical field. Their poor stability is yet another major barrier to the use of these compounds. As a result, they are vulnerable to degradation brought on by a variety of factors, including high oxygen levels. The phytosome technology meets this challenge by markedly enhancing the bioavailability of phytoconstituents as well as their stability because, when entrapped in a carrier, they become protected from degradation and oxidation [19,20]. Phytosomes are formed as a result of the stoichiometric complexation of phospholipids with polyphenolic constituents such as flavonoids, phenolic acids, tannins, and terpenoids within an organic solvent. The hydroxyl groups of phytoconstituents form hydrogen bonds with amphiphilic lipids such as phospholipids and cholesterol and form lipid-compatible supramolecular adducts with enhanced stability and bioavailability [21].

The objective of the present study was to evaluate the efficacy of the optimized nanophytosomal gel prepared from the dried hydroalcoholic extract of *S. platensis* (SPNP-gel) on the wound healing of injured rats. This was conducted by investigating the autophagy process (LC3B/Beclin-1), NRF-2/HO-1 antioxidant pathway, inflammatory factors (TNF-, NF-κB, TlR-4, and VEGF), apoptotic processes (Apoptosis-inducing factor, AIF, and Caspase-3), and the HGMB-1 protein expression in comparison to the marketed product for wound healing and plain *S. platensis* gel.

## 2. Results

### 2.1. Determination of Total Phenolic Content of Algal Extract

The total phenolic content (TPC) of the lyophilized powder of algal extract was expressed in terms of GAE (Gallic Acid Equivalent). The regression equation obtained from the standard plot of gallic acid was used to calculate the TPC. It was estimated by 37.225 ± 1.16 mg GAE/g extract.

### 2.2. Molecular Networking (MN) and MRM Methodology

MN is used to display parent ions in tandem mass spectrometry (MS/MS) based on their fragmentation pattern similarities. The resulting network of *S. platensis* extract is shown in Figure 1. The molecular nodes and clusters were further annotated, compared with public libraries available on the GNPS platform, and matched with the reference compounds available. All annotated compounds that matched with references are indicated in green in Figure 1. Nodes that belong to the same cluster have similar fragmentation patterns and are linked by a cosine score equal to or >0.6. The degree of quality of these libraries and the reliability of annotations are defined as either gold, silver, or bronze based on the quality of the matched reference spectra. Gold spectra express structurally characterized synthetic or purified compounds. On the other hand, silver reference spectra must be supported by a publication, while bronze spectra are all the other remaining putative annotations. The quality of these reference spectra matched compounds found in *S. platensis* extract are labeled with gold, silver, and bronze coins, as shown in Figure 1. Accordingly, the largest clusters found were clusters of amino acids and proteins such as phenylalanine, tyrosine, etc. Two nodes of one nucleotide and one nucleoside were annotated, representing guanosine monophosphate and deoxyguanosine, respectively. Moreover, one cluster contained pterin, a known biomolecule synthesized by *S. platensis* [22]. Another cluster represented a node with an oxylipin, 12,13-dihydroxy-9Z-octadecenoic acid (12,13-DiHOME) [23]. In addition, several nodes contained known acids such as pantothenic acid (vitamin B5), 16-hydroxypalmitic acid (a fatty acid), azelaic acid, and ophthalmic acid (a glutathione analog) [24]. Loliolide, a naturally occurring benzofuran, was also identified according to the GNPS library [25].

Polyphenol identification in the hydroalcoholic extract of *S. platensis* was achieved by LC-MS based on MRM transitions (see Section 4.2.5). According to the work of Mapoung et al. (2020) [26], *S. platensis* was found to contain many phenolic acids in addition to catechin and quercetin. So, a standard mixture of 12 phenolic acids (chlorogenic acid, caffeic acid, gallic acid, syringic acid, protocatechuic acid, p-Coumaric acid, ferulic acid, vanillic acid, benzoic acid, 4-hydroxy benzoic acid, cinnamic acid, and salicylic acid) and two flavonoids (catechin and quercetin) was prepared and injected according to the procedure mentioned in Section 4.2.5. Three criteria were considered: 1/the retention time, 2/the molecular ion, and 3/MRM transitions. The first transition corresponded to the most abundant fragmentation ion, while the second transition was specific to the analyte and was used as a qualifier for polyphenols. Four phenolic acids were identified: salicylic acid, cinnamic acid, benzoic acid, and 4-hydroxy benzoic acid (Table 1). After identification, the standards of the four phenolic acids were re-injected into LC-HRMS (see Section 4.2.6), then added to the MN shown in Figure 1 and labeled with pink.

### 2.3. Docking Study

The molecular docking of all 13 identified compounds on the active site of the HMGB-1 protein showed comparable affinities for the identified compounds towards the protein, as shown in Table 2. The highest docking score of −7.12962 kcal/mol was shown by 12,13-DiHome, followed by 16-hydroxypalmitic acid with a docking score of −6.89744 kcal/mol. The interaction of 12,13-DiHome with the active site of the HMGB-1 protein was established through ARG23, LYS67, and LYS11 with the two oxygen atoms of the carboxylic acid functional groups via hydrogen bonding and ionic interactions. Interaction distances ranged between 2.9 and 3.03 Å, with estimated binding energies of −5 kcal/mol for ionic interaction and −15.9 kcal/mol for hydrogen bonding. Additionally, there is hydrophobic affinity through PHE37, VAL19, PHE40, VAL35, and TYR15 (Figure 2).

### 2.4. Analysis of 22 Full Factorial Design of the Developed Nanophytosomal Dispersions

The developed formulae were optimized using 2^2^ full factorial designs (Design-expert^®^ software version 7, Stat-Ease, Inc., Minneapolis, MN, USA). Four experimental runs were generated by the software for the development of nanophytosomes (Table 3). Two-factor interaction (2FI) was the chosen model. To ensure that the model could be used to navigate the design space, an adequate precision ratio was chosen. A ratio larger than four is preferred, and this was recorded for all dependent variables (Table 3). The predicted and adjusted R^2^ values were acceptable for all dependent variables.

### 2.5. Study of the Effect of Formulation Variables on EE%, PS, ZP, PDI, and Q6h of the Developed Nanophytosomes

Considering both the PC amount (X_1_) and CH amount (X_2_), it was found that they influenced all investigated dependent variables (Table 4 and Figure 3). By increasing both PC and CH amounts, EE% and PS values increased (*p* < 0.0001). In addition, it is well known that the influence of cholesterol on EE% and PS was greater than that of PC, as F3 (PC:CH 50:50) was remarkably greater in EE% and PS than that of F2 (PC:CH 100:0). Similarly, PDI increased with lipid content and ranged from 0.43 ± 0.06 to 0.69 ± 0.13, which indicates that the prepared nanophytosomes were polydisperse but with acceptable values. ZP of all nanophytosomes were negative and ranged from −11.13 ± 0.20 to −19.8 ± 0.49 mV. However, no correlation was found between ZP value and lipid amount when lipid content was 100 mg and below, but a significant increase in ZP absolute value was noticed for total lipid content of 150 mg. Regarding Q6h values, it is worth noting that increasing the amount of lipid content resulted in a decrease in the Q6h values with no notable difference between F2 and F3, which have the same lipid content but different ratios of PC and CH.

### 2.6. Selection of the Optimum Nanophytosomal Formulation

Using Design Expert^®^ software, numerical analysis was used to determine the desired level of variables to obtain *S. platensis* nanophytosomes with the highest EE% and ZP (absolute value) and the lowest PDI, PS, and Q6hrs. Accordingly, (F4), which included 100 mg PC and 50 mg CH (Figure 4), was selected for further investigations.

### 2.7. Study of the Morphology of the Optimum Nanophytosomal Formulation (F4) Using TEM

The morphological study of F4 revealed that the vesicles had a homogeneous size distribution and a spherical shape (Figure 5). Zetasizer’s PS estimate for the vesicles was in good agreement with the findings of TEM.

### 2.8. FTIR Study

The complex formation between the components of the phytosome (PC and CH) and *S. platensis* was identified and characterized using FTIR. The FTIR spectra of dried *S. platensis* extract, PC, CH, and nanophytosomes are illustrated in Figure 6. The FTIR spectrum of *S. platensis* extract exhibited a broad IR band from 3493 to 3255 cm^−1^ due to acidic carboxylic groups and other characteristic bands at 3134 (H on Sp^2^), 2925 (H on Sp^3^), 1671 (acidic -C=O group), and 1599 and 1541 cm^−1^ (aromatic ring). PC showed characteristic peaks at 3363, 2927.93 and 2861.81 cm^−1^, as well as several sharp peaks below 1800 cm^−1^. The FTIR spectra of spirulina–loaded nanophytosomes showed the disappearance of the band due to carboxylic groups in *S. platensis* extract and the lack of the functional group (O-H) of CH at 2936 cm^−1^. Additionally, a disappearance of the PO_4_ band at 1371 cm^−1^ in the FTIR spectrum of the PC and the appearance of a new band at 1735 cm^−1^ in the phytosomal formulation could be observed. The new band at 1735 cm^−1^ is probably due to the C=O stretching of the ester linkage of phospholipids. All of the above confirm the successful formation of the complex.

### 2.9. Stability Study

The visual examination of the stored system did not show any change in the physical appearance, sedimentation, or aggregation throughout the storage time. Moreover, the values of EE%, ZP, PS88, and PDI for the stored optimum formula were 61.55 ± 0.99, −20.00 ± 2.00 mV, 602.10 ± 2.nm and 0.600 ± 0.001, respectively. No statistical difference was found in EE%, ZP, PS, and PDI (*p* > 0.05 for all values) between the stored and freshly prepared formulae. Furthermore, the pH value of the dispersion (6.8 ± 0.02) did not change during storage.

### 2.10. Characterization of the SPNP-Gel

The optimized formula was incorporated into a 2% HPMC gel (SPNP-gel). The gel revealed optimum results for topical application to open wounds with respect to organoleptic properties, pH, drug content, rheological behavior, and spreadability. The detailed characterization of the gel is provided in the Appendix A.

### 2.11. Effect of SPNP-Gel Topical Application on Wound Contraction Rate

To assess the healing potential of nano-spirulina (SPNP)-gel after wound induction, the contraction rate was observed and documented by capturing images at different time intervals (0, 3, 6, 9, 12, and 14 days). The analysis revealed that MEBO^®^ ointment (GpIII/standard group; injured rats in this group were treated topically with MEBO^®^ 0.2 g two times/day), spirulina gel (GpIV; rats in this group were treated topically with spirulina platensis gel as the standard group), and SPNP gel groups (GpV; rats in this group were treated topically with SPNP-gel as the standard group) markedly increased the contraction rate as compared to the +ve control group (wounded, untreated rats) starting from day 3 after surgery. Furthermore, the wound-contraction rate of SPNP-gel was significantly higher than that of other treatment groups at all the time intervals evaluated.

The comparison of healing power between the SPNP-gel-treated group and other treatment groups revealed that on day 3, all showed the same effect, while on days 6 and 9, the effect of SPNP-gel on the contraction rate in group VI seemed to be greater than that in other treatment groups (GPIII and GPIV). Moreover, on day 12, partial closure of wounds was observed only in the SPNP-gel-treated group, and other treated groups showed decreased wound size without complete wound closure. Finally, on the last day of the investigation (day 14), complete closure of the wound and a normal appearance of the skin were observed only in the rats treated with SPNP-gel (Figure 7).

### 2.12. Effect of SPNP-Geltopical Application on High Mobility Group Box-1 (HMGB-1) Expression, and Inflammatory Markers in Wounded Rats

The +ve control rats demonstrated a significant increase in (A&A*) HMGB-1 protein expression that, in sequence, increased (B) toll-like receptor (TLR-4) and (C) nuclear factor Kappa B (NF-κB) levels as compared to the normal healthy one (Figure 8). However, rats treated with MEBO^®^ ointment (standard) or *S. platensis* gel downregulated HMGB-1 expressions, showing anti-inflammatory potential affirmed by decreasing serum inflammatory biomarkers as compared with wounded rats. Moreover, the post-treatment of wounded rats with SPNP-gel showed superior modulation to HMGB-1, increasing its anti-inflammatory potential compared to other treatment groups (Figure 8).

Consequently, the vast increase in inflammatory markers increased immune-histochemistry expression of TNF-α in the control +ve group compared to the skin of normal rats. In contrast, all the post-treatment groups presented a significant reduction in TNF-α positive staining. The slightest TNF-α reading was detected in the standard treated group, followed by that in the SPNP-gel group (Figure 9).

### 2.13. Effect of SPNP-Gel Topical Application on Anti-Oxidative Stress Markers in Wounded Rats

As presented in Figure 8, wounded rats showed a significant decrease in levels of (D) nuclear factor erythroid-2 (Nrf-2) and (E) heme oxygenase-1 (HO-1) when compared to the negative control group. Moreover, the post-treatment with MEBO^®^ and *S. platensis* gel indicated a noticeable rise in Nrf-2 and HO-1 levels when compared to wounded rats (+ve control group). Furthermore, SPNP-gel showed a higher antioxidant sequel prior to its application, which was relevant to the outcomes of the other treated groups.

### 2.14. Effect of SPNP-Gel Topical Application on Autophagy Markers in Wounded Rats

As shown in Figure 10, rats with untreated wounds presented a marked decrease in the autophagy process, as evidenced by low (10A&A*) light chain 3 (LC3) and (10B&B*) Beclin-1 protein expressions as compared to the intact skin of the control negative group. On the other hand, all rats treated with MEBO^®^ or *S. platensis* gel showed enhancement in both autophagy-marker expressions. Notably, the groups treated with SPNP-gel showed a marked increase in LC3BII/I and Beclin-1 markers compared to the other treated groups’ results.

### 2.15. Effect of SPNP-Gel Topical Application on Apoptotic Markers in Wounded Rats

As shown in Figure 10, the untreated wounded rats (+ve control) demonstrated a significant increase in (C) caspase-3 levels and (D&D*) apoptosis-inducing factor (AIF) expression as compared to the intact skin of normal rats. However, all the treatment regimens showed anti-apoptotic effects by reversing these effects. Further, the SPNP-gel showed the highest antiapoptotic activity among other treatment regimens (Figure 10).

### 2.16. Effect of SPNP-Gel Topical Application on the Percentage Change of Vascular Endothelial Growth Factor (VEGF) in Wounded Rats

The positive expression of VEGF (Figure 11) markedly increased in all treated groups when compared to the +ve control group. Moderate VEGF expression was noticed in the *S. platensis* gel group, while the highest value of VEGF was noticed in the standard group, followed by the SPNP-gel-treated group.

### 2.17. Effect of SPNP-Gel Topical Application on Histopathological Alterations and Collagen Formation in Wounded Rats

As shown in Figure 12 and Figure 13, a microscopic examination of the skin from the -ve control group revealed normal histology of skin composed of epidermis and dermis that was rich in collagen with the presence of numerous hair follicles, sweat, and sebaceous glands. On the contrary, the wound area in the +ve control group showed signs of retarded healing; despite filling the wound gap with organized tissue, the absence of partial re-epithelialization was noticed. The inflammatory reaction was marked, and the angiogenesis was poor. The standard-treated group showed marked improvement; the wound surface was covered by a thick and complete layer of newly formed epithelium, and the wound gap was filled with collagen-rich fibrovascular organized tissue. A minimal inflammatory reaction was noticed in the wound area. *S. platensis* gel enhanced the process of wound healing as re-epithelialization was perfect, and the inflammation subsided to a great extent. Good quality organized tissue was noticed in the examined wound sections. Better healing criteria were detected in SPNP-gel treated group, as the wound area was contracted with perfect closure and complete re-epithelialization.

Generally, all treated groups exhibited significantly higher scores in healing criteria when compared to the +ve control group. A significantly higher score of re-epithelialization was detected in both standard and SPNP-gel-treated groups without a significant difference in between. A significantly higher granulation tissue score was detected in the standard group compared to the SPNP-gel group. A significant reduction in inflammation was noticed in both the standard and SPNP-gel groups with the absence of significance. The angiogenesis score in the SPNP-gel group was comparable to that detected in the standard group.

Concerning collagen content in the wound area (Figure 13), the +ve control group showed a significant reduction in collagen content in the wound area when compared to the -ve control and other treated groups. All treated groups showed a significant increase in collagen content in the wound area. The highest significant increase in collagen was detected in both the standard and SPNP-gel-treated groups. No statistically significant difference was noticed between the standard and SPNP-gel-treated groups.

## 3. Discussion

Finding a novel, cheap, and convincing treatment strategy is essential to fast-tracking the wound healing and regeneration process with minimal complications. An unhealed wound is highly liable to infectious microorganisms, especially resistible ones. Untreated wounds may disrupt internal organ functions, resulting in instantaneous death in severe cases or immunocompromised patients. Lately, there has been a high tendency worldwide to use natural resources and reveal their mechanistic cues targeting the complicated wound healing process and reactions inside the body [27].

Some of the phytocompounds have proven to be highly effective in wound healing because they were found to enhance wound closure by stimulating the growth factors involved in the healing process, regulating collagen synthesis, and promoting its deposition in the wound bed [28]. In the current study, 13 compounds could be identified, including phenolic acids, fatty acids, alcohols, nucleoside derivatives, and others. The molecular docking of the identified compounds on the active site of HGMB-1 was studied to reveal their effect in downregulating the expression of the HGMB-1 protein, which played a prominent role in wound healing (see Section 2.3). The effect of these compounds on the reduction of HGMB-1 protein expression could not be found in the literature. However, the contribution of most of them to wound healing by other mechanisms is reported.

In the extract, a few phenolic acids were found, namely cinnamic acid, 4-hydroxybenzoic acid, salicylic acid, and benzoic acid, with docking scores of −4.458, −4.296, −4.631, and −4.233 kcal/mol on the active site of HMGB1 protein, respectively (Table 2). It is stated by Viana et al. (2021) that trans-cinnamic acid regulates wound healing by increasing the migration of fibroblasts [29]. In addition, Sinha et al. (2019) tested the dermal anti-inflammatory activity of salicylic acid-loaded nanoemulsion and could show that the developed formula considerably diminished the topically induced inflammation in mice by inhibiting the expression of pro-inflammatory cytokines and lowering the production of COX and LOX [30]. Moreover, the activation of nuclear factor-κB, which is a rapid-response transcription factor playing an important role in stress responses and inflammatory reactions, can be inhibited by salicylates. This has great value in the management of hypertrophic scarring [31]. El-Zaway et al. (2022) discovered that benzoic acid could fight pathogens involved in burn wound infection, and it could be a hopeful candidate for wound healing [32]. Although no data could be found in the literature documenting a wound-healing effect for 4-hydroxybenzoic acid, the current study explores its wound-healing activity with a docking score of −4.458 kcal/mol on the active site of HMGB1 protein.

In the *S. platensis* extract loliolide, which is a monoterpenoid hydroxy lactone, was identified with a docking score of −4.699 kcal/mol. A study conducted by Park et al. (2019) investigated the antiapoptotic and wound-healing effects of loliolide using HaCaT cells (a human keratinocyte cell line) [25]. They found that the expression of caspases (3, 8, and 9), related to ROS-induced apoptosis, was reduced by loliolide. In addition, loliolide improved the expression of the keratinocyte growth factor and inflammatory cytokines, such as interleukin (IL)-1, IL-17, and IL-22, as well as the epidermal growth factor receptor signaling pathway (PI3K, AKT) and migration factors, such as K6, K16, and K17. This indicates that loliolide has the potential to heal wounds.

Pantothenic acid (vitamin B5) could also be identified in the *S. platensis* extract, which was found to have a docking score of −5.330 kcal/mol. The effect of pantothenic acid on wound healing was previously investigated on fibroblastic cell cultures, and over the first few days, a considerable rise in cell proliferation and 3H-thymidine incorporation was observed [33]. Even when pantothenic acid was given as a supplement to rabbits to investigate its effect on wound healing, a significantly greater fibroblast content was observed during the fibroblast proliferation phase [34].

According to the GNPS library, azelaic acid could also be identified as having a docking score of −5.372 kcal/mol, which indicates its prominent contribution to the wound-healing effect of the *S. platensis* extract through HMGB1 reduction. In addition, it is reported that azelaic acid inhibits toll-like receptor 2, decreases expression of kallikrein-5 (KLK-5) and pro-inflammatory cathelicidins such as LL-37, and aids in the removal of reactive oxygen species (TLR-2) [35].

Ophthalmic acid, also found in the *S. platensis* extract, is a precursor of gluthaione [36], which was reported to reduce oxidative stress, and to reestablish the MMP-1/TIMP-1 ratio, giving way to adequate and regular extracellular matrix production and re-epithelialization [37].

Deoxyguanosine, which was identified in the *S. platensis* extract with a docking score of −5.839 kcal/mol, belongs to the class of organic compounds known as purine 2′-deoxyribonucleosides. Chen et al. (1999) investigated the effect of a topically applied mixture of deoxyribonucleosides, including deoxyguanosine, on the acceleration of wound healing in an open wound model in rabbits [38]. They found a remarkable increase in total new granulation tissue and a significantly higher incidence of complete wound re-epithelialization compared to controls.

Guanosine-5′-monophosphate, a purine ribonucleoside monophosphate, showed a relatively high docking score (−6.502 kcal/mol). Korotkina et al. (1980) reported that cyclic guanosine monophosphate could accelerate wound healing via the elevation of gluconeogenesis [39].

Moreover, this study also explored a wound-healing activity for pterin (docking score −4.572 kcal/mol) as well as for 16-hydroxypalmitic acid (docking score −6.897 kcal/mol). Chuah et al. (2018) reported that palmitic acid could not facilitate wound healing. Although it showed a significant effect in downregulating the pro-inflammatory cytokines (TNF- and IL-12), it could not enhance the proliferation of fibroblasts and collagen synthesis [40].

The highlight of the current study is the discovered wound healing ability of 12,13-DiHome, which revealed a docking score of −7.930 kcal/mol indicating that the action is mediated through the downregulation of HMBG-1 protein. It is worth noting that no wound healing activity is previously documented in the literature for pterin, 16-hydroxypalmitic acid or for 12,13-DiHome.

A modern approach for improving the efficacy of phytocompounds at the skin level is represented by their incorporation into phospholipid vesicles, especially phytosomes, owing to their biocompatibility and similarity to skin components, which facilitates their interaction with the skin tissue in addition to being accumulated in deep skin layers. In the current investigation, nanophytosomes loaded with dried extract of *S. platensis* were developed and characterized, and the optimized formula was formulated as a gel (SPNP-gel) to facilitate its topical application on open wounds.

It is well known that the synthesis of phytosomes is based on a stoichiometric reaction involving the formation of hydrogen bonds between the phytochemicals having an active hydrogen atom (e.g., -COOH, -OH, -NH2, -NH, etc.) and the hydrophilic parts of the amphiphilic molecules. Each amphiphilic molecule, either PC or CH, may form a complex with one or two of the phytochemicals. Therefore, it is expected that with increased lipid content, the EE% and, accordingly, the PS will increase, as logically, the number of complexed molecules will increase. The increased EE% of F3 over F2 indicates that CH complexes contain more phytomolecules than PC. The complex formation could be evidenced by comparing the FTIR spectra of pure CH, pure PC, and the dried *S. platensis* extract with those of the developed phytosome (Figure 6). The disappearance of the band characteristic for the carboxylic groups in *S. platensis* extract as well as the O-H stretching band of CH in the FTIR spectrum of the phytosomes indicate the formation of the complex between cholesterol and the extracted phytoconstituents. Furthermore, the disappearance of the PO_4_ band in the FTIR spectrum of the PC and the appearance of a new band in the phytosomal formulation suggest that the functional groups in *S. platensis* interacted with the phosphate group of the PC by hydrogen bonding in the *S. platensis* phytosome. This interaction explains the observed high-loading value of *S. platensis*-extracted phytoconstituents. These results are consistent with those obtained by Faezeh et al. (2022), who confirmed the physicochemical interaction between caffeine and chlorogenic acid (5-caffeoylquinic acid) in coffee silverskin and PC by FTIR, paying attention to significant differences between the pure compound and phytosome complex [41]. Additionally, it was observed that the sharp endothermic character of the functional groups in the spectra of coffee silverskin-enriched extract and phospholipid had been significantly suppressed. This result is consistent with the current FTIR spectrum’s verification.

The remarkable increase in PS of the formulae containing CH, namely F3 and F4, may also be attributed to the fact that including cholesterol into the bilayer alters the lipid vesicles’ geometrical packaging, which would accordingly affect lipid vesicle size, the curvatures, and rigidity of the surface bilayer [42,43]. Furthermore, Sułkowski et al. (2005) stated that when cholesterol is incorporated in the vesicular membrane, the distance between the phospholipid chains will increase and the possibility of interaction between the electronic shells of phospholipids’ polar head groups in the bilayer will become limited [44]. The change in the geometry of the vesicles due to CH and accordingly the increased size might have affected the particles’ dispersibility, leading to higher PDI values. Knowing that PC and CH are neutral, the negative charge acquired by the nanophytosomes is probably due to the weak acidic character of the encapsulated polyphenols. The increased absolute ZP of F4 in comparison to the other formulae is probably attributed to the changed three-dimensional geometry of the nanovesicles due to increased lipid content, which may orient more negative charges at the surface of the vesicles. The Q6h values are in great accordance with the EE% data. The lower the EE% of the encapsulated compound, the greater the free unencapsulated part, which results accordingly in a higher amount of released drug. No remarkable difference could be detected between Q6h values of F2 and F3, despite the higher EE% and greater PS of the latter, which would both result in a significantly lower release due to a reduced fraction of free drug and a smaller surface area, respectively. In addition, the presence of cholesterol in F3 would reduce fluidity and microviscosity, preventing leakage and hence reducing the membrane permeability of water-soluble molecules [45]. This finding could possibly be attributed to the fact that CH forms a weaker bond with polyphenols than PC. According to the desired criteria, which are the highest EE% and ZP (absolute value) and the lowest PDI, PS, and Q6h, F4 was selected as the optimum formulation and was incorporated in a gel preparation for the in vivo study.

A well-documented wound excision model was used for wound injury induction in healthy adult male Sprague-Dawley rats to track the modulatory potential of the developed formulation. Rats topically treated with SPNP-gel for two weeks showed higher wound closure potential as compared to standard and self/physiologically healed rats. SPNP-gel promoted the wound healing process and enhanced skin morphology visually and microscopically through the downregulation of high mobility group box-1 (HGMB-1) protein expression, enhancing the collagen production, autophagy process (light chain 3 (LC3)B/Beclin-1), antioxidant pathway (nuclear factor erythroid-2 (NRF-2)/Heme Oxygenase-1(HO-1)), and vascular endothelial growth factor (VEGF) expression, halting the inflammatory [tumor necrosis factor alfa (TNF-α), nuclear factor kappa B (NF-κB), toll-like receptor-4 (TLR-4)), and apoptotic processes (apoptosis inducing factor (AIF), Caspase-3).

HGMB-1 is a highly mobile non-histone nuclear protein that controls gene transcription and modulates various critical physiological processes in the body [46,47]. HGMB-1 is implicated in the pathogenesis of many disorders other than dermatological ones, such as ischemic, neurodegenerative, autoimmune, and arthritic diseases. Its role in skin disorders is not fully illustrated and still requires more investigation to be fully clarified [48]. In the current study, untreated wounded rats presented a mountainous increase in HGMB-1 protein expression as compared with the ones with normal intact skin. In line with our findings, a recent clinical study documented a high level of HGMB-1 in the plasma of patients with acne vulgaris [48]. On the other side, wounded rats treated for 14 days with SPNP-gel not only reduced HGMB-1 but also abated the results of the other treatment regimens, providing a faster wound closure process with a higher contraction rate. A previous study agrees with our findings, where wound closure was enhanced by reducing HGMB-1 levels in the incisional rat wound model [49]. Furthermore, C-phycocyanin, one of *S. platensis* microalgae’s active constituents, was documented to reduce HGMB-1 protein expression in a gastric ulcer rat model. Hence, SPNP-gel’s ability to modulate HGMB-1 increased its merit.

As further documentation, molecular docking was applied to all 13 identified compounds on the active site of the HMGB-1 protein, showing comparable affinities for the identified compounds towards the protein HMGB-1 secondary structure, which showed 6-α helices, 5-β turns, 2-γ turns with 67% α-helices, 32% coil, and 9% turn without β-sheet, and classified as an α-class protein. The prominent active site has residues: Tyr78, Ile79, Pro80-81, Lys82, Gly83v, Glu84, Thr85, Lys86-88, Phe89, Lys90, Asp91, Pro92, Asn93, Tyr162, and Lys165 with 310 Å3 site volume. The interacting residues of the CGA-HMGB-1 docked complex were ILE79, PRO80-81, LYS82, GLY83, GLU84, LYS86-88, PHE89, Arg163, Ala164, LYS165, and Gly166 [50]. The identified compounds of the *S. platensis* extract showed high affinities for interaction with the HMGB-1 protein, as previously discussed in Section 4.1.

Escalated inflammatory processes account for wound-associated complications such as scarring and even cancer in chronic, unhealed wounds [51]. In the present study, a marked inflammatory response following wound excision is documented by the increase in TNF-α, NF-κB, and TlR-4 protein expression of the skin tissue of wounded rats compared to normal ones. The acute inflammatory response in the wound environment involves the activation of pro-inflammatory cytokines, such as TNF-α, which further mediates NF-κB signaling via toll-like receptors [52], which supports our results. Moreover, HGMB-1 up-regulation may play a part in the inflammatory response observed in wounded rats, as it is considered a potent proinflammatory cytokine that triggers the buildup of cytokines, increases chemotaxis, and stimulates many cells involved in the inflammatory process, such as immune and endothelial cells, as well as fibroblasts. In contrast, all the treatment regimens showed anti-inflammatory activity by reducing TNF-α, NF-κB, and TlR-4 expressions compared to the untreated group. SPNP-gel-treated rats presented the highest anti-inflammatory effect attributable to the higher modulatory capacity of HGMB-1.

Increased controlled oxidative stress and redox signaling regulate the proper wound healing process by aiding hemostasis, inflammation, angiogenesis, wound closure, and extracellular matrix. In contrast, uncontrolled ones play a central role in the pathophysiology of chronic unhealed wounds by maintaining and deregulating inflammatory processes [53,54]. Heme oxygenase-1 (HO-1) has a vital role in cytoprotection against oxidative stress via its antioxidant enzymatic activity. The transcription of HO-1 is mediated by Nrf-2 by nuclear translocation after disconnection from its cytoplasmic inhibitor, which causes stimulation of several detoxifying genes and antioxidant enzymes, such as HO-1 [55]. Furthermore, delayed wound healing was associated with the suppression of Nrf-2 signaling and its downstream HO-1. In this investigation, wound excision hampered the Nrf-2/HO-1 cue as compared to non-injured rats. An impaired antioxidant system increases oxidative stress and inflammatory cytokine production; hence, targeting and constructing a formula with antioxidant potential is highly needed [56]. The topical application of SPNP-gel showed the highest antioxidant capacity, among treatment groups, by upsurging Nrf-2/HO-1 protein expressions compared to normally healed rats, and this antioxidant power was well documented by previous studies [57,58].

Autophagy is a well-conserved recycling mechanism that is required to ensure the survival and maintenance of cells. Any disturbance in the autophagy process, either increased or decreased, involve in many diseases’ progression. Concerning skin and wound healing, this process is intricate and vague, and targeting it aids in treatment solutions, especially refractory ones [59]. In the current work, the autophagy process was investigated by measuring the LC3BII/I ratio and Beclin-1, where rats with untreated wounds presented a decreased LC3BII/I ratio and Beclin-1 levels reflected an inhibition in the autophagy process compared to normal healthy rats. Treatments with SPNP-gel showed the best enhancement in the autophagy process among treatment groups (standard and *S. platensis* gel) by increasing the LC3BII/I ratio and Beclin-1 levels as compared to untreated wounded rats. According to our knowledge, there is no data on spirulina’s effect on autophagy in skin models. However, *S. platensis* was documented to protect the liver by enhancing the autophagy process against high-energy diets in sheep [60]. In agreement with our data, rapamycin regulated the proliferation and emigration of fibroblasts from damaged wounds by intensifying the autophagy process [61].

An additional cellular process that is required to maintain cellular homeostasis and stability is apoptosis, which is highly connected to the autophagy process, regardless of the significant differences between them [62]. Apoptosis is highly embroiled in the initial phases of normal wound healing for the removal of inflammatory cells and scar creation with minimal tissue damage [63]. However, the increased apoptotic process has been associated with abnormal wound healing responses, such as scarring of the injured area. During apoptosis, executioner caspases mediate cell death [14,64]. Caspase-3 is one of the primary apoptotic executors, responsible for the proteolytic cleavage of numerous important proteins, which are cleaved in a variety of systems during apoptosis. Furthermore, AIF, which is a mitochondrial oxidoreductase that contributes to programmed cell death in a caspase-independent manner [65], was investigated for the importance of caspase-independent cell death trajectories in many pathologies/diseases [66]. Wounded rats showed a considerable increase in AIF and caspase-3 as compared to unwounded rats, demonstrating a high evolution of the apoptotic process. The SPNP-gel application showed the highest antiapoptotic activity among other treatment groups. This effect was apparent in the morphological appearance of the skin, where this group showed no scarring and the best wound closure. The SPNP-gel’s antiapoptotic potential is a consequence of its higher anti-inflammatory and antioxidant potential. The verity that apoptosis is modulated by oxidative stress and inflammation was established in many studies with different pathologies [67]. Further, its capability to downregulate HMGB-1 also played a part in this inhibition [46,47]. Finally, enhancing autophagy by increasing beclin-1 inhibited the apoptotic process, where it was previously documented that enhancing autophagy in the early stages of brain injury after a subarachnoid hemorrhage protected neurons against neural apoptosis through a mitochondrial pathway [38]. Neovascularization is momentous in the wound healing process, which is controlled by several factors such as hypoxia, inflammation, and growth factors. Strategies involving the enhancement of growth factors in the wound area are emerging. VEGF is one of the angiogenic growth factors that promote endothelial cell proliferation and microvascular permeability and controls numerous endothelial integrin receptors during blood vessel sprouting [68,69]. Additionally, VEGF-interrupted expression was reported after wound injury and during the healing process [70], which is in accordance with our data. On the other side, the different treatments terminated these disturbances in VEGF in immunostained tissue, with MEBO showing the highest increase, followed by nanoform, and finally *S. platensis* gel. The increase in VEGF was reported to enhance wound healing [70] and that was apparent herein from skin morphology, wound contraction rate, and histopathological findings. Additionally, the topical application of VEGF accelerated healing and contraction in refractory wounds such as diabetes [71]. Ultimately, the upregulation of endogenous VEGF by SPNP-gel depicted the survival of skin tissue by increasing its antioxidant, anti-inflammatory, and anti-apoptotic promises [68,69]. Collagen, as a main component of the extracellular matrix, possesses an essential regulatory role in wound healing phases, and any disturbance in collagen may result in a chronic, resistant state [72]. In this work, all the treated groups have significantly reverted the wound effect on collagen. Further, an enhancement of wound re-epithelialization was associated with increased collagen expression relative to the wounded untreated group that was highlighted previously [72]. Strengthening this potential, many wound dressing formulations containing collagen or collagen enhancers are available on the market [73]. Therefore, the ability of SPNP-gel to enhance collagen production added a higher value to its use in this milieu.

## 4. Materials and Methods

### 4.1. Chemicals and Reagents

L-α phosphotidylcholine (PC), cholesterol (CH), and methanol (70%) were purchased from Sigma-Aldrich (St. Loius, MO, USA). Hydroxypropyl methyl cellulose HPMC K4M (colorcon, Kent, UK). MEBO^®^ ointment (Golf Pharmaceutical Industrials, Ras Al-Khaimah, UAE) was purchased from the local market (Cairo, Egypt). MEBO^®^ ointment’s active constituent is B-sitosterol (0.25%), and the base is composed mainly of sesame oil and beeswax. BG11 media (medium for blue-green algae) was provided by Thermo Fisher Scientific (Waltham, MA, USA). All other reagents were of analytical grade.

### 4.2. Methods

#### 4.2.1. Isolation and Purification of *S. platensis*

Algal species were isolated from River Nile water and concentrated using a phytoplankton net (80 µm mesh size) and BG11 media. BG11 media (500 mL) containing 1.5% agar were spread in Petri dishes, and water samples of 0.1 mL were inoculated in the media for the isolation of the algal strain. *S. platensis* colonies were transferred in a liquid medium (50 mL) mounted with white fluorescent illumination (intensity x˜2500Lu) for 24 h at ambient temperature.

#### 4.2.2. Cultivation of the Isolated Strains

Cultivation was conducted under continuous illumination in BG11 (600 mL) in sterile 1-L conical shoulder flasks at 25 °C. Growth rate was monitored until reaching a plateau where the culture reached maximum growth and became suitable for harvest (15–20 days).

#### 4.2.3. Preparation of Algal Extract

The dried algal biomass (500 gm) was extracted by maceration using methanol (70%) in an ultrasonic water bath at 40 °C. The extract was filtered on a 0.45 µm membrane filter, and the filtrate was dried using a freeze-drying apparatus (Alpha-1-2-LDplus, Donau Lab, Ukraine) set at atmospheric pressure in warm-up mode for 11 min and at 0.37 mbar in main drying mode for 4 hrs. The yield product (7.7 g), a dark, amorphous, sticky mass, was kept in a dark vial at a temperature of −20 °C.

#### 4.2.4. Estimation of Total Phenolic Content of *S. platensis* Algal Extract

The Folin–Ciocalteu method for the estimation of total phenolic content was adopted [74]. In this method, gallic acid solution was used as a standard. A sample of the algal extract was prepared in concentrations of 1 mg/mL in 100% methanol. Gallic acid (1 mg) was dissolved in 100% methanol (stock solution) and used for serial dilutions in the range of 1000–50μg/mL. One mL of Folin–Ciocalteu phenol reagent was added, and the mixtures were incubated at room temperature for 30 min. The results were recorded using the microplate reader FluoStar Omega (BMG LABTECH, Ortenberg, Germany) as each of the 7 standards and 1 sample was pipetted into the plate wells in 6 replicates. Measuring the absorbance against the reagent blank was performed at 630 nm. Total phenolic content was assessed as gallic acid equivalent in mg (GAE).

#### 4.2.5. Liquid Chromatography Coupled to Mass Spectrometry (LC-MS)

Polyphenols separation was achieved by using UHPLC-MS/MS (Thermo Fisher Scientific, Les Ulis, France) according to a published method [75] with an Agilent Zorbax SB-C18 column (Torrance, CA, USA) (100 mm × 2.1 mm × 1.8 μm) heated in a heating chamber at 40 °C. Elution was achieved by a binary mixture of solvents (A–B). Solvent A was water acidified with 0.1% formic acid, and solvent B was acetonitrile containing 0.1% (*v*/*v*) of formic acid. The flow rate of the mobile phase was 0.4 mL/min using the following gradient: 10–18% B (0–1 min), 18–33% B (1–6.5 min), 33% B (6.5–9.5 min), 33–40% B (9.5–15 min), 40–90% B (15–16 min), 90% B (16–19 min), and 90–10% B (19–20 min). Mass spectrometry analysis was achieved on an Agilent 6430 triple quadrupole mass spectrometer (Torrance, CA, USA). Electrospray ionization (ESI) was used as the ionization source with the following parameters: 11 l/min of drying gas (nitrogen), 15 psi nebulizer pressure, 350 °C temperature, and 3000 V capillary voltage. The detection of targeted analytes was performed using a multiple reaction monitoring mode (MRM). The retention time and two fragmentation ions were the two parameters used for the identification (quantifier and main qualifier transitions) and optimized with pure standards.

#### 4.2.6. Liquid Chromatography Coupled to High Resolution Mass Spectrometry (LC-HRMS)

The preparation of the sample was conducted according to the following procedure: 50 mg of the lyophilized powder of the hydro-alcoholic extract of *S. platensis* was reconstituted in 1 mL of a solvent mixture of water: methanol: acetonitrile 50/25/25 (*v*/*v*). The sample was mixed for 2 min using a vortex and then sonicated for 10 min by an ultrasonicator, followed by centrifugation at 10,000 rpm for 10 min. Fifty µL of stock solution was diluted to 1000 µL by reconstitution solvent. Accordingly, the injected concentration was 2.5 mg/mL.

Separation was realized using a Vanquish Flex system (Thermo Fisher Scientific, Illkirch-Graffenstaden, France) consisting of an autosampler, a binary pump, and a heated compartment. A Luna Omega PS C18 100 A column (150 mm × 2.1 mm, 5 μm) from Phenomenex (Torrance, CA, USA) was used for the analysis. The flow rate was 500 μL/min and the injection volume was 5 µL. The used gradient elution is composed of 0.1% formic acid in water as solvent (A) and 0.1% formic acid in acetonitrile as solvent (B). Solvent B was varied as follows: 0 min, 1%; 11.5 min, 40%; 14 min, 95%; 14.5 min, 1%; and 16 min, 1%. The temperature of the column was kept at 25 °C.

A Q-Exactive mass spectrometer equipped with a heated electrospray ionization (HESI-II) probe (Thermo Fisher Scientific, Illkirch-Graffenstaden, France) was used for the MS detection. A weekly-based calibration for the mass analyzer was achieved using Pierce^®^ ESI Positive and Negative Ion Calibration Solutions (Thermo Fisher Scientific, Illkirch-Graffenstaden, France).

The parameters of HRMS were adjusted as follows: spray voltage at 4 kV; capillary temperature and probe heater temperature at 200 °C; sheath gas at 60 a.u.; auxiliary gas at 20 a.u.; capillary temperature and probe heater temperature at 200 °C; and S-lens RF level at 50. A full-scan MS mode in the range of m/z 70–1050 was selected for the acquisition at a resolution of 140,000 FWHM. The maximum injection period was 100 ms, and the automated gain control (AGC) target was 3 × 106 ions. Data-dependent acquisition mode (dd-MS2) was used to record the generated spectra at a resolution of 17,500 FWHM. The Xcalibur Qual Browser version 2.1 (Thermo Fisher Scientific, Illkirch-Graffenstaden, France) was used to process the data.

#### 4.2.7. Molecular Networking and Annotation of Molecules

An MSConvert program 64 bit (https://proteowizard.sourceforge.io/download.html accessed on 6 January 2020) was used to convert the data acquired on a Thermo Scientific HRMS instrument to Mzmine compatible format (i.e., Mzml). The data were then processed and visualized with Mzmine 2.53 (MZmine3Download) to obtain a Global Natural Product Social Molecular Networking (GNPS) compatible format. The results were then exported to a molecular networking (MN) analysis on a GNPS site (GNPS—Analyze, Connect, and Network with your Mass Spectrometry Data (ucsd.edu)). The following parameters were considered: precursor ion and fragment ion mass tolerance were set both at 0.02 Dalton, minimum pair cosine was 0.6, network TopK was 10, and maximum connected component size was 100. Results obtained after the MN analysis on GNPS were processed by the Cytoscape program 3.8.2 (Cytoscape 3.8.2 Release Note) to visualize the fragmentation patterns acquired by HRMS in a ddMS2 mode.

#### 4.2.8. Docking Study

Using the Discovery Studio tool (version 2021), 3D structures of the compounds found in the bioactive extract were constructed, establishing a database that was then used as an input file for DockingServer. Docking calculations were performed using DockingServer [76]. Ligand atoms were prepared for docking through the addition of gaseous partial charges and non-polar hydrogen. Single bonds were allocated, and all spatial confirmations were identified.

Virtual docking of ligand atoms on the active site of the HMGB1 protein structure was performed. AutoDock tools (version 4.2.6.) were used for the addition of essential hydrogen atoms, Kollman-type charges, and dissolution conditions [77], as well as the calculations of van der Waals forces, electrostatic energies, and distance-dependent dielectric functions. Affinity (grid) maps were created by the Autogrid tool.

Docking simulation was performed using the Lamarckian Genetic Algorithm (LGA) and the Solis and Wets local search method [78]. The original location, positioning, and torsion of the ligand molecules were set in a random manner. During docking, all rotatable torsions were released. Each docking process was developed from 2 separate runs, each of which was adjusted for a limit of 250,000 energy estimations. The population size was set at 150 at the maximum. A translational step of 0.2 Å, quaternion step of 5, and a torsion step of 5 were included in the search criteria.

#### 4.2.9. Preparation of Nanophytosomes Loaded with Dried *S. platensis* Extract

Nanophytosomes were assembled by applying the thin-film hydration technique using PC and CH at different amounts (Table 4). At first, in a round flask, PC, CH, and dried *S. platensis* extract (50 mg) were dissolved using 10 mL methanol. A thin film of phytosomes was formed by evaporating the organic phase under vacuum at 60 °C and 90 rpm for 30 min, utilizing a rotatory evaporator (Rotavapor, Heidolph VV 2000, Burladingen, Germany). The film hydration was conducted using glass beads for 45 min in 10 mL distilled water at 60 °C, and the pH of the dispersion was measured using a pH meter (JENWAY model 350, JENWAY Ltd., London, UK). Afterward, the *S. platensis* nanodispersions were kept at 4 °C.

#### 4.2.10. Entrapment Efficiency Percentage Determination (EE%)

Using a cooling centrifuge, the produced nanodispersions were centrifuged at 20,000 rpm for 1 h at 4 °C (Sigma 3 K 30, Osterode am Harz, Germany). The sediments were then lysed with methanol and subjected to a UV-Vis spectrophotometer analysis for the detection of *S. platensis* phytoconstituents at a maximum wavelength of 273 nm (Shimadzu UV1650 Spectrophotometer, Kyoto, Japan). Each measurement was made three times with a standard deviation. The following equation was used to determine the EE%:$$\mathrm{EE}\%=\left(\frac{Entrapped\;phytoconstituents\;concentration}{Total\;phytoconstituents\;concentration}\right)\times 100$$

#### 4.2.11. Particle Size, Zeta Potential Determination of Formulated Nanophtosomes

A Malvern Zetasizer 2000 (Malvern Instruments Ltd., Malvern, UK) was used to measure the mean particle size (PS), zeta potential (ZP), and polydispersity index (PDI) of the diluted nanophytosomal dispersions in deionized water (1:100).

#### 4.2.12. Determination of Percentage of Drug Released after 6 h (Q6h)

The USP dissolution apparatus (Pharma Test, Hamburg, Germany) was used for performing the in vitro release experiment after being modified by removing the baskets and replacing them with cylindrical plastic tubes with one end connected to the shafts and the other end covered by cellulose membrane (12,000–14,000 molecular weight cutoff). Two milliliters of nanophytosomal dispersions were put in each cylinder to test the drug release for 6 h at 37 °C. The reservoir medium was 50 mL of PBS with a pH of 5.5 and aliquots were removed every hour and replaced with fresh medium. A spectrophotometrical analysis was conducted for the samples at λ_max_ 273 nm. The measurements were carried out in triplicates and represented as mean ± SD.

#### 4.2.13. Experimental Design and Selection of the Best Achieved Nanophytosomal Formula

Utilizing Design Expert^®^ software version 7, a full factorial design (2^2^) was created to assess the effect of several factors on nanophytosomal formulation (Stat Ease, Inc., Minneapolis, MN, USA). Four runs were required by the design. Two factors were chosen as independent variables, namely PC amount (X_1_) and cholesterol amount (X_2_). The dependent variables were EE% (Y_1_), PS (Y_2_), PDI (Y_3_), ZP (Y_4_), and Q6h (Y6) (Table 2). Afterward, the optimum formula was selected using the desirability function, which allows the simultaneous investigation of all variables. The selection of the formula with the best characteristics relied on having the smallest PS, PDI, and Q6h and the highest EE% and ZP as absolute values. The suggestion with the highest desirability solution (near to one) was opted (Table 3). The selected formula was subjected to subsequent characterization.

#### 4.2.14. Fourier Transform-Infrared Spectroscopy (FTIR)

The FTIR spectrum of the lyophilized optimum *S. platensis* nanoparticle formula, SPNP (F4), was recorded in comparison with that of the lyophilized *S. platensis* extract, CH, and PC using an FTIR spectrophotometer (FTIR spectrophotometer, Model Genesis II Mattson, Fremont, CA, USA). A homogeneous mixture of the sample and KBr was formed and compressed into discs; the FTIR spectra were traced in the region of 400–4000 cm^−1^ [79].

#### 4.2.15. Stability Study of Optimum Nanophytosomal Dispersion (SPNP)

The physical stability testing of SPNP was performed to examine any physical changes that may take place during storage. SPNP was stored at 4 °C for 1 year then the pH, PS, PDI, EE%, ZP, and Q6h were measured and the received data were compared with that of the freshly prepared formulation. The results were statistically analyzed using Student’s *t*-test (SPSS^®^ 25.0 software). The differences were considered significant at *p* ≤ 0.05. The system was also visually examined for any signs of particle aggregation or sedimentation.

#### 4.2.16. Morphology Study of Optimum Nanophytosomal Dispersion

The morphology of optimum nanophytosomal dispersion was examined using transmission electron microscopy (TEM; Joel JEM 1230, Tokyo, Japan). A thin film of the sample was placed on a carbon-coated copper grid and stained using phosphotungstic acid 1.5% for TEM examination.

#### 4.2.17. Formulation of SPNP Gel and *S. platensis* Gel

To facilitate the application of the formula to the open wound, an HPMC gel was prepared. The *S. platensis* extract and the optimum *S. platensis* loaded nanophytosomal formula (SPNP) were mixed with HPMC using a magnetic stirrer to prepare a 2% w/w *S. platensis* gel and an SPNP-gel, respectively. The detailed characterization of the gel is provided in the Appendix A.

#### 4.2.18. Creation of Excision Wound and Animal Distribution

A total of 40 adult male Sprague-Dawley rats (250–270 g) were habituated individually in plastic cages to minimize biting and possibly wound scraping by other rats and to prevent contamination following injuries throughout the trial. They were also maintained at a constant room temperature (20–25 °C) and fed a regular diet.

On day 0, the day of wound creation, all rats were anesthetized with halothane (1.5%) and an intraperitoneal injection of pentobarbital (0.5 mL/kg). All the rats’ backs were shaved and cleaned with 70% ethanol, and a 10 mm-diameter circular, full-thickness skin wound was created in the center of each animal’s back, with the exception of the rats in the control-negative group [80].

Randomly, all animals were distributed over 5 groups: *n* = 8: GPI/negative (-ve) control: non-wounded rats; GpII/positive (+ve) control: wounded rats; GpIII/standard group: wounded rats that were treated topically with a commercial cream in the local market for wound healing (MEBO^®^) (0.2 g 2 times/day for 2 weeks) [81]; GpIV/*S. platensis* gel: rats in this group were treated topically with *S. platensis* gel (0.2 g 2 times/day for 2 weeks); and GpV/SPNP-gel: rats in this group were treated topically with SPNP-gel (0.2 g 2 times/day for 2 weeks). All the treatments started 24 h after wound creation. Wound contraction was assessed visually and digitally captured using a digital camera (Canon Power Shot S200, Tokyo, Japan) at 0, 3, 6, 9, 12, and 14 days following wound creation. The institutional ethics committee at Misr University for Science and Technology for the use of laboratory animals accepted the protocol of this study (reference number: PT14, 14/3/2022) that is adhered to the Guides for the Care and Use of Laboratory Animals (NIH publication, 1996).

#### 4.2.19. Preparation of Blood and Tissue Samples

Prior to scarification and the excision of the wound peripheral tissue area on day 14, each rat had an eye puncture into serum separator tubes after their euthanasia. The serum was then collected, spun at 3000 rpm for 15 min, and stored at −80 °C until the biochemical parameters were analyzed. The collected tissues were divided into two halves. One half was stoked for western blot analysis at −80 °C, and the other half was retained in 10% neutral buffered formalin for the histopathological study.

#### 4.2.20. Determination of Serum Parameters

ELISA kits were handled in accordance with the manufacture’s directions to measure serum levels of toll-like receptors-4 (TLR-4; Cusabio Biotech Co., Wuhan, China; Cat. # CSB-E15822r) and nuclear factors kappa B, nuclear factor erythroid-2, heme oxygenase-1, and caspase-3 (NF-κB, Nrf-2, HO-1; MyBioSource, CA, USA, Cat. # MBS015549, MBS752046, MBS764989, and MBS7244630, respectively).

#### 4.2.21. Western Blotting

The following parameters were determined using the Western blotting assay: HMGB-1, AIF, LC3BII/I, and Beclin-1 (primary antibodies were purchased from Thermo Scientific Co., IL, USA), and the steps adhered to the formerly mentioned technique by El Gazar et al. (2022) [82]. Specific bands were visualized with a ChemiDocTM imaging system (Image LabTM software version 5.1, Bio-Rad Laboratories Inc., Hercules, CA, USA). The optical density (OD) of the following findings was standardized against β-actin.

#### 4.2.22. Histopathological Evaluation

On day 14, the skin tissue samples that were collected after animals’ scarification and stabilized in 10% neutral buffered formalin were immersed in melted paraffin wax after being processed in different grades of alcohol and xylene. Sections of 5µm each were cut and stained with hematoxylin and eosin (H&E). For collagen assessment, tissue sections were stained with the Masson trichrome stain (MTC) [83]. The wound healing criteria were evaluated as described formerly by Bakr et al. (2021) [84]. Briefly, re-epithelialization was given a number ranging from 0 to 4 describing the degree and quality. Granulation tissue formation was evaluated on a scale from 0 to 4 to describe the degree of organization. The degree of inflammation was given a number from 0 to 4 to describe the reduction in the number of inflammatory cells per microscopic field. Angiogenesis was also evaluated microscopically by the number of vessels present per site.

#### 4.2.23. Immunohistochemistry Study

As formerly described by Gendy et al., (2022) [85], 5 µm slices were sliced into positive charged slides, rehydrated, and heat-retrieved before being incubated with primary anti-TNF- and VEGF for the duration of the night (at a dilution of 1:100). Following washing, tissue sections were incubated for 30 min at room temperature with a 1:1000 dilution of an HRP-labeled secondary antibody before being blocked for endogenous peroxidases. The color was created using a DAB-Substrate Kit. Slides with negative controls were produced by skipping the primary antibody stage. Using an Olympus BX43 microscope, slides were inspected, and an Olympus DP-27 camera was used to take pictures (Tokyo, Japan). Using Cell Sens Dimensions, positive immune staining was measured as an area percentage (Olympus software).

#### 4.2.24. Statistical Analysis of In Vivo Study

All data were presented as mean values with standard deviation (SD). A one-way analysis of variance (ANOVA) was used to assess the data, followed by Tukey’s post hoc test. A statistically significant difference was defined as a *p*-value less than 0.05.

## 5. Conclusions

In the current study, 13 phytoconstituents could be identified in the hydroalcoholic extract of *S. platensis* using LC-MS, LC-HRMS, and molecular networking. Optimized nanophytosomes of lyophilized *S. platensis* extract were developed and formulated as an HPMC gel (SPNP-gel). The wound healing potential of SPNP-gel was investigated on injured Sprague-Dawley rats in comparison to the marketed wound healing product (a standard, MEBO^®^ ointment) and a gel prepared from *S. platensis* extract (*S. platensis* gel). The SPNP-gel showed superior wound healing potential among treatment groups through the downregulation of HGMB-1 protein expression. A docking study of the identified phytoconstituents on the active site of the HGMB-1 protein showed a significant effect of the extracted compounds on the reduction of HGMB-1 levels. The docking score of −7.129 kcal/mol recorded for 12,13-DiHome indicates its prominent contribution to wound healing potential of the *S. platensis* extract. Further, the developed formula showed remarkable anti-inflammatory activity by reducing TNF-α, NF-κB, and TLR-4 expressions, as well as the highest antioxidant capacity by upsurging Nrf-2/HO-1 protein expressions. Moreover, treatment with SPNP-gel showed the best enhancement in the autophagy process among treatment groups by increasing the LC3BII/I ratio and Beclin-1 levels. It also revealed the highest antiapoptotic activity by showing no scarring and the best wound closure among all treated injured rats. In addition, it showed a significant increase in VEGF, which is crucial for angiogenesis. Finally, the developed formula showed a great enhancement of wound re-epithelialization that was associated with increased collagen expression. Collectively, SPNP-gel resulted in significantly better wound closure, a higher contraction rate, increased collagen production, and enhanced histopathology at the site of injury when compared to *S. platensis* gel and showed comparable or even superior results in many investigated parameters when compared to the standard. This finding indicates the remarkable superiority of the nanoform over the ordinary extract in reaching the target cells and exerting the action. Consequently, the present study proves that formulating phytoconstituents in the form of nanophytosomes significantly improves their effectiveness. Relocating this study to clinical trials, especially in resistant and refractory ones, is highly recommended.

## Figures and Tables

**Figure 1 marinedrugs-21-00149-f001:**
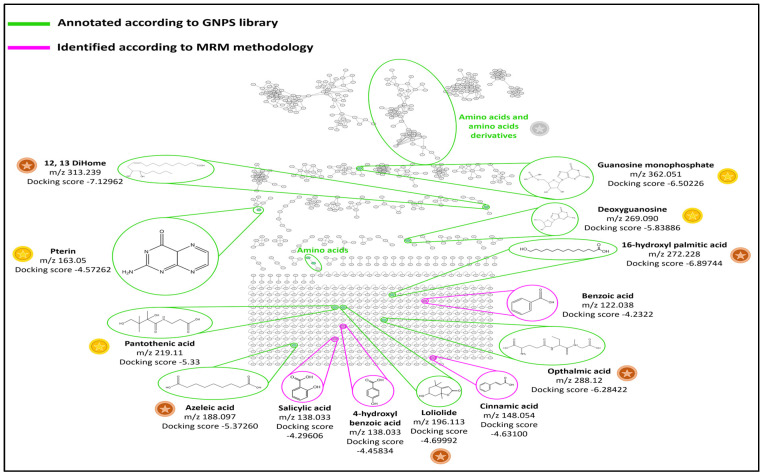
Cytoscape view of the entire molecular network and nodes of major compounds annotated by GNPS with their docking score of *S. platensis* hydroalcoholic extract on the active site of HMGB1 protein expressed as the estimated free energy of binding in kcal/mol. The green color represents compounds annotated according to the GNPS library. The pink color represents compounds identified according to the MRM methodology. Quality of the GNPS reference spectra, gold (
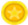
), silver (
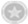
), and bronze (
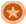
).

**Figure 2 marinedrugs-21-00149-f002:**
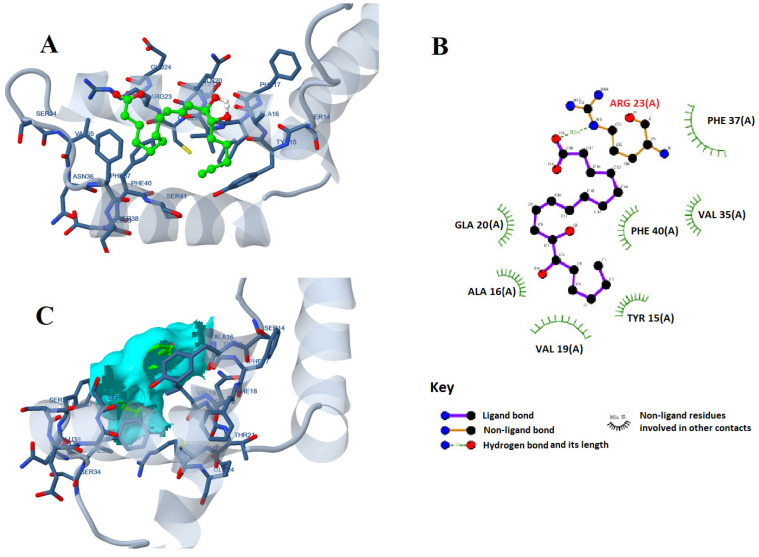
The 2D (**A**) and 3D (**B**) interactions of 12,13-DiHome with the active site of (**C**) the surface area of the interaction of the ligand atom interaction with nucleic/amino acids of the HMGB1 protein.

**Figure 3 marinedrugs-21-00149-f003:**
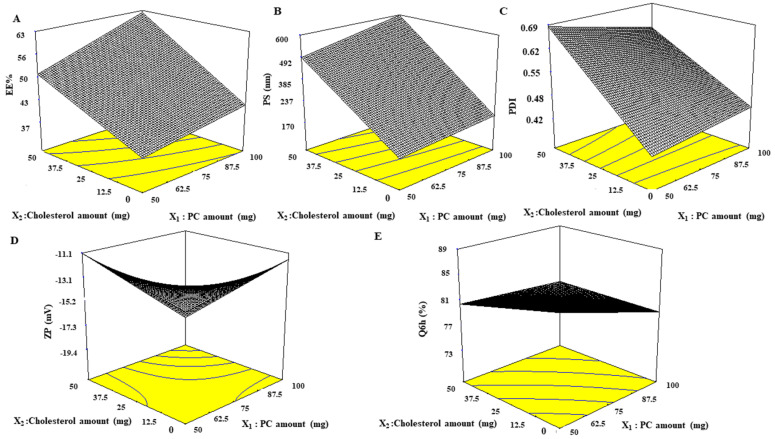
Response 3D plots for the effect of the PC and CH amounts on (**A**) EE%, (**B**) PS, (C) PDI, (**D**) ZP, and (**E**) Q6h of nanophytosomes. Abbreviations: PC—phosphatidylcholine, CH—cholesterol, EE%—entrapment efficiency percentage, PS—particle size, PDI—polydispersity index, ZP—zeta potential, and Q6h—amount released after 6 h.

**Figure 4 marinedrugs-21-00149-f004:**
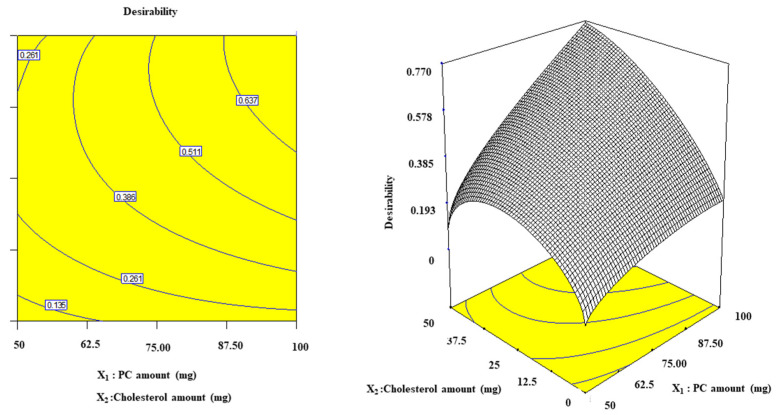
The 3D desirability diagram of optimization conducted by using the Design-Expert^®^ V7 program for the determination of the optimum nanophytosomal formula.

**Figure 5 marinedrugs-21-00149-f005:**
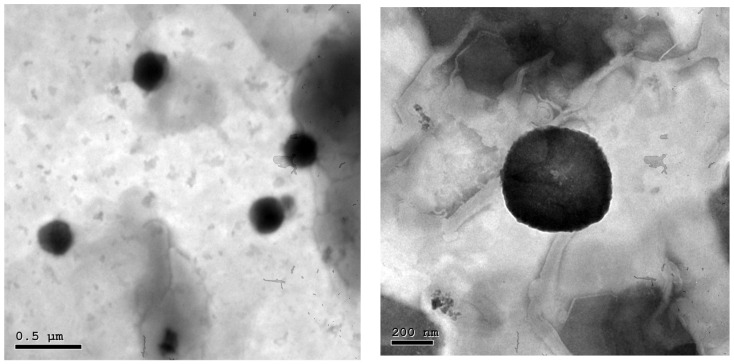
Transmission electron micrographs of an optimized nanophytosome (F4) at different magnifications.

**Figure 6 marinedrugs-21-00149-f006:**
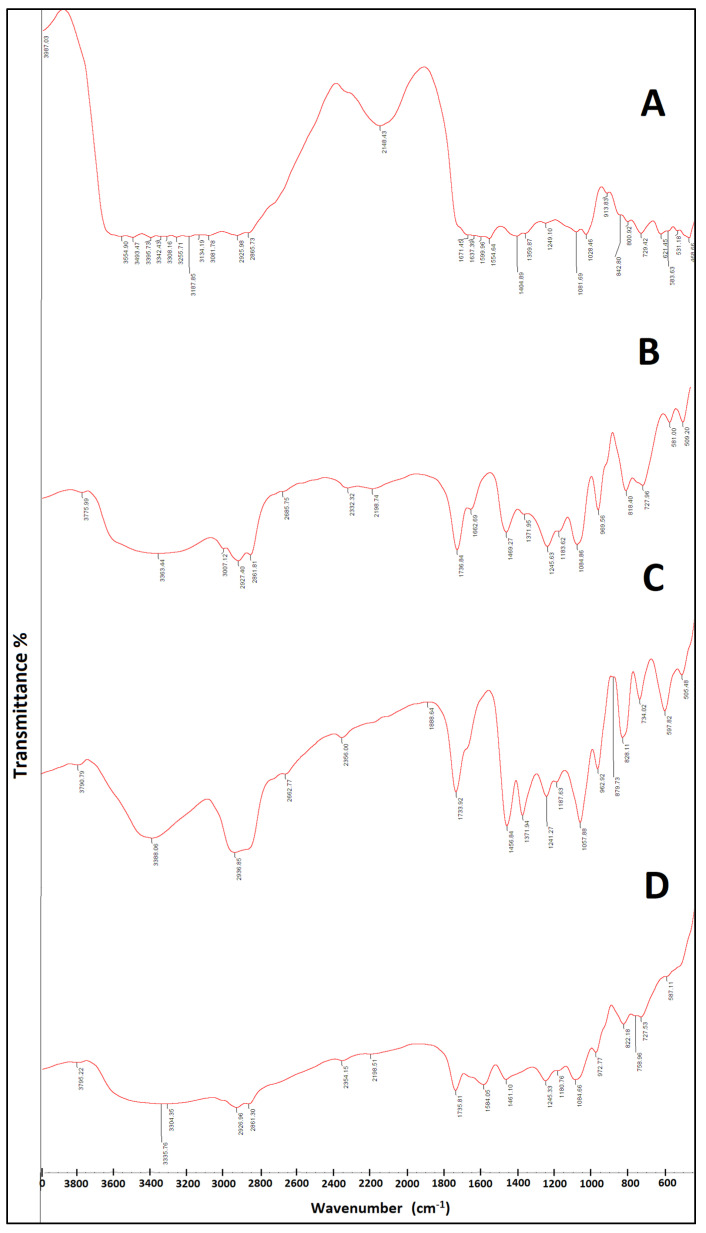
The FTIR spectra of (**A**) *S. platensis* extract, (**B**) phosphatidylcholine (PC), (**C**) cholesterol (CH), and (**D**) optimized nanophytosome (F4).

**Figure 7 marinedrugs-21-00149-f007:**
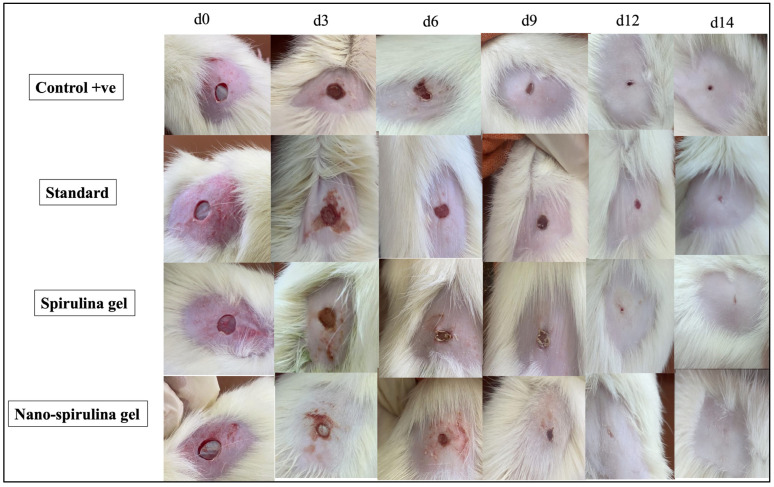
Representative images monitoring rats’ skin after wound excision and a thorough healing process on days 0, 3, 6, 9, 12, and 14 with the standard marketed product (MEBO^®^), *S. platensis* gel, and nano-*S. platensis* gel (SPNP-gel) in comparison to +ve control. Note: GpII/control positive (+ve): untreated wounded rats; GpIII/standard group: wounded rats treated with MEBO^®^; GpIV/*S. platensis* gel: wounded rats treated with *S. platensis* gel; and GpV/nano-spirulina (SPNP)-gel: wounded rats treated with SPNP-gel.

**Figure 8 marinedrugs-21-00149-f008:**
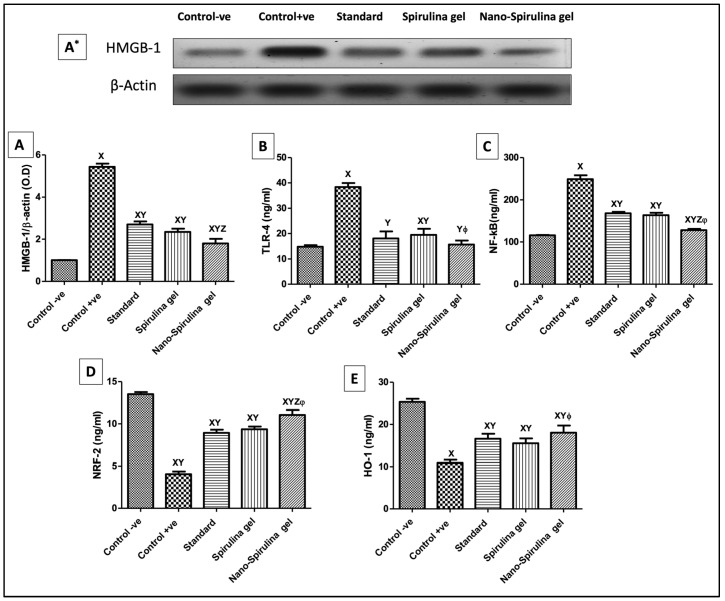
Changes in skin protein expression/contents of (**A**,**A***) HMGB-1, (**B**) TLR-4, (**C**) NF-κB, (**D**) Nrf-2, and (**E**) HO-1 due to the application of standard, *S. platensis* gel, and nano-*S. platensis* gel (SPNP-gel); **A*** is the Western blot result for HMGB-1. The values are presented as mean (*n* = 8) ± SD and a statistical analysis was carried out using a one-way ANOVA, followed by Tukey’s post hoc multiple comparison test. As compared with control -ve (X), control +ve (Y), standard (Z), *S. platensis* gel (ϕ); *p* < 0.05. Abbreviations: HO-1—heme oxygenase-1, HMGB-1—high mobility group box-1, Nrf-2—nuclear factor erythroid-2, NF-κB—nuclear factor Kappa B, and TLR-4—toll-like receptor. Note: GpI/control negative (-ve): non-wounded rats; GpII/control positive (+ve): untreated wounded rats; GpIII/standard group: wounded rats treated with MEBO^®^; GpIV/*S. platensis* gel: wounded rats treated with *S. platensis* gel; and GpV/nano-spirulina (SPNP)-gel: wounded rats treated with SPNP-gel.

**Figure 9 marinedrugs-21-00149-f009:**
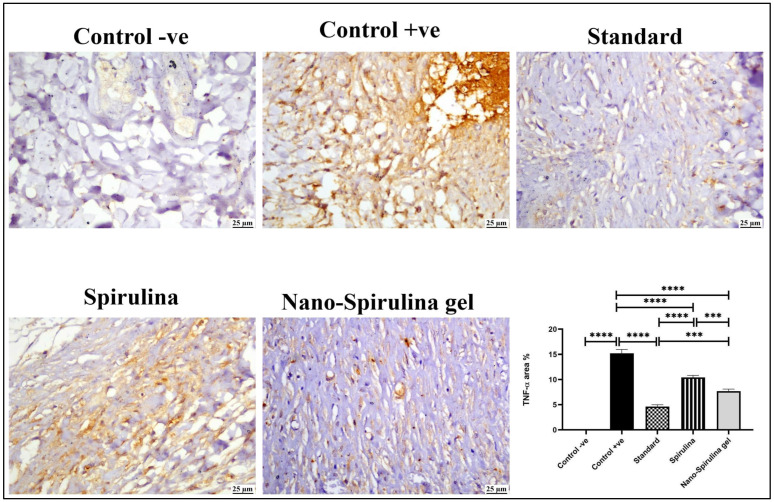
Photomicrograph of skin (immune staining) showing TNF-α expression at the wound area, control -ve showed negative expression of TNF-α, control +ve group showed marked TNF-α positive staining, mild TNF-α expression was noticed in standard and nano-*S. platensis* gel (SPNP-gel) groups and moderate expression was noticed in the *S. platensis* gel group. The chart describes the quantification of TNF-α positive staining as area percent, data were presented as means ± SD, and a significant difference was considered at *p* ˂ 0.05. Note: GpI/control negative (-ve): non-wounded rats; GpII/control positive (+ve): untreated wounded rats; GpIII/standard group: wounded rats treated with MEBO^®^; GpIV/*S. platensis* gel: wounded rats treated with *S. platensis* gel; and GpV/nano-spirulina (SPNP)-gel: wounded rats treated with SPNP-gel. Number of asterisks “*” above the columns indicates strength of significance.

**Figure 10 marinedrugs-21-00149-f010:**
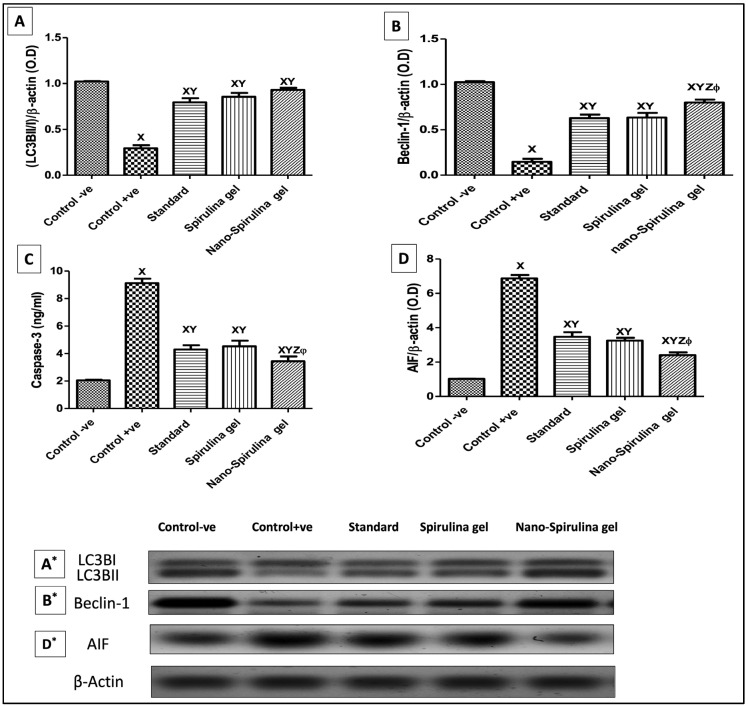
Changes in skin protein expression/contents of (**A**,**A***) LC3BII/I, (**B**,**B***) Beclin-1, (**C**) Caspase-3, and (**D**,**D***) AIF due to application of standard, *S. platensis* gel and nano-*S. platensis* gel (SPNP-gel); **A***, **B***, and **D*** are the Western blot results for LC3BII/I, Beclin-1, and AIF, respectively. The values are presented as mean (*n* = 8) ± SD and a statistical analysis was carried out using a one-way ANOVA followed by Tukey’s post hoc multiple comparison test. As compared with control -ve (X), control +ve (Y), Standard (Z), *S. platensis* gel (ϕ); *p* < 0.05. Abbreviations: LC3B—light chain 3B and AIF—apoptosis-inducing factor. Note: GpI/control negative (-ve): non-wounded rats; GpII/control positive (+ve): untreated wounded rats; GpIII/standard group: wounded rats treated with MEBO^®^; GpIV/*S. platensis* gel: wounded rats treated with *S. platensis* gel; and GpV/nano-spirulina (SPNP)-gel: wounded rats treated with SPNP-gel.

**Figure 11 marinedrugs-21-00149-f011:**
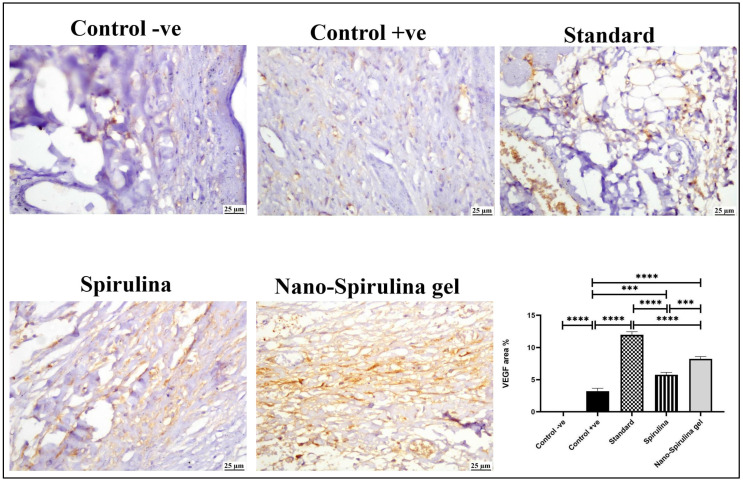
Photomicrograph of skin (immune staining) showing the VEGF expression at the wound area; control -ve showing negative expression of VEGF; control +ve group showing very limited negative VEGF staining; moderate VEGF expression was noticed in the *S. platensis* gel group; and marked increase in VEGF was noticed in the standard and nano-*S. platensis* gel (SPNP-gel) groups. The chart describes the quantification of VEGF positive staining as area percent, data were presented as means ± SD, and a significant difference was considered at *p* ˂ 0.05. Note: GpI/control negative (-ve): non-wounded rats; GpII/control positive (+ve): untreated wounded rats; GpIII/standard group: wounded rats treated with MEBO^®^; GpIV/*S. platensis* gel: wounded rats treated with *S. platensis* gel; and GpV/nano-spirulina (SPNP)-gel: wounded rats treated with SPNP-gel. Number of asterisks “*” above the columns indicates strength of significance.

**Figure 12 marinedrugs-21-00149-f012:**
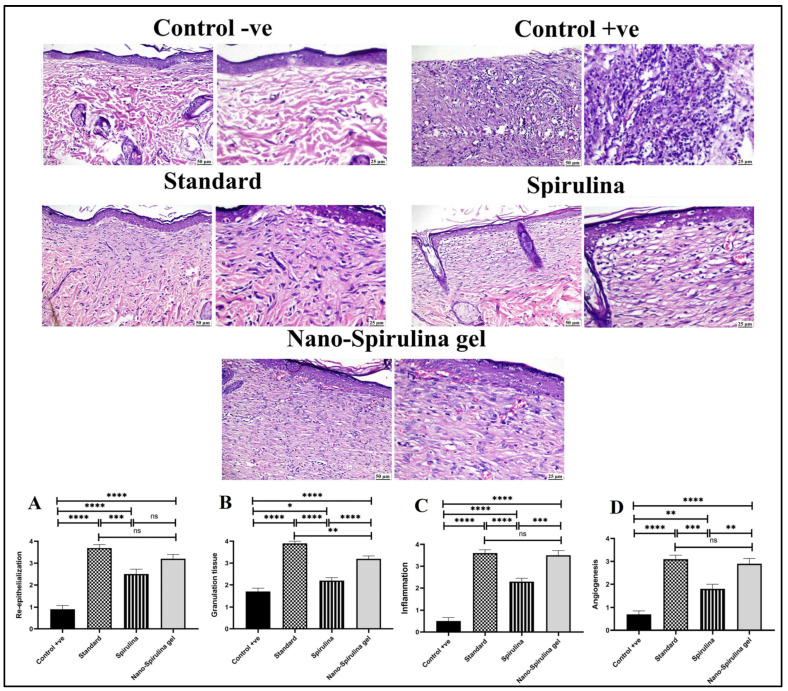
Photomicrographs of skin (H&E) stained showing normal skin in the -ve control group, the absence of epithelial cover and intense inflammation in the +ve control group, good re-epithelialization and collagen-rich granulation tissue in the standard group, complete re-epithelialization, good granulation tissue formation and mild inflammation in the *S. platensis* gel group, perfect wound closure, re-epithelialization, and minimal inflammation in the nano-*S. platensis* gel (SPNP-gel) group. The charts describe the histopathological scores of wound healing criteria in different experimental groups. (**A**) Re-epithelization score, (**B**) Granulation tissue formation, (**C**) Inflammation score and (**D**) Angiogenesis. Control -ve group showed normal skin (no healing criteria could be evaluated). Data were presented as means ± SD, and a significant difference was considered at *p* ˂ 0.05. Note: GpI/control negative (-ve): non-wounded rats; GpII/control positive (+ve): untreated wounded rats; GpIII/standard group: wounded rats treated with MEBO^®^; GpIV/*S. platensis* gel: wounded rats treated with *S. platensis* gel; and GpV/nano-spirulina (SPNP)-gel: wounded rats treated with SPNP-gel. Number of asterisks “*” above the columns indicates strength of significance, (ns) means non-significant.

**Figure 13 marinedrugs-21-00149-f013:**
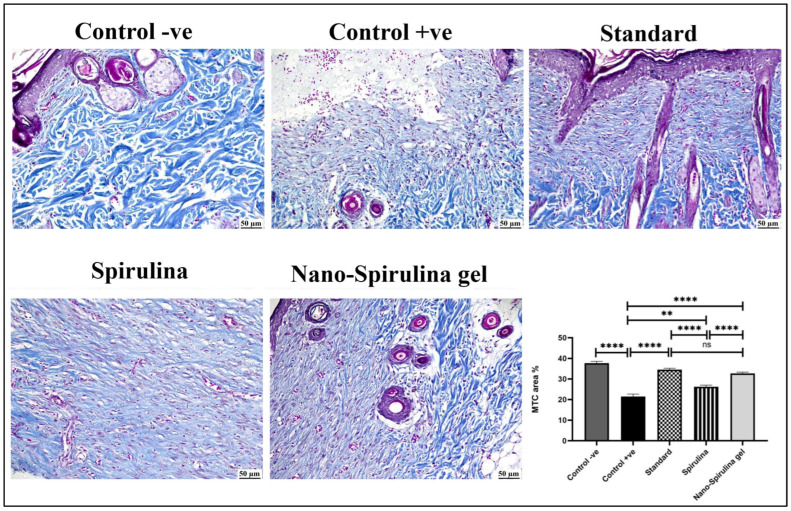
Photomicrograph of skin (MTC) stained showing high collagen content (blue color) in the -ve control group, decreased collagen content in the +ve control group, increased collagen in the standard group, a moderate increase in collagen in the *S. platensis* gel group, and markedly increased collagen in the nano-*S. platensis* gel (SPNP-gel) group. The chart describes the amount of collage expressed as area percent of blue color, data were presented as means ± SD, and a significant difference was considered at *p* ˂ 0.05. Note: GpI/control negative (-ve): non-wounded rats; GpII/control positive (+ve): untreated wounded rats; GpIII/standard group: wounded rats treated with MEBO^®^; GpIV/*S. platensis* gel: wounded rats treated with *S. platensis* gel; and GpV/nano-spirulina (SPNP)-gel: wounded rats treated with SPNP-gel. Number of asterisks “*” above the columns indicates strength of significance.

**Table 1 marinedrugs-21-00149-t001:** Retention time and MRM transitions of polyphenols.

Polyphenols	Retention Time (min)	Molecular Weight	Transition Time 1 (*m*/*z*)	Transition Time 2 (*m*/*z*)
Salicylic acid	8.17	138	139 → 95	139 → 77.1
Cinnamic acid	19.22	148	149 → 131	149 → 103
Benzoic acid	15.02	122	123 → 79.1	123 → 77.1
4-hydroxybenzoic acid	15.61	138	139 → 65	139 → 39

**Table 2 marinedrugs-21-00149-t002:** Docking score of identified compounds of *S. platensis* on the active site of HMGB1 protein expressed as the estimated free energy of binding in kcal/mol.

Compound	Docking Score kcal/mol
12,13-DiHome	−7.12962
16-Hydroxypalmitic acid	−6.89744
Guanosine-5′-monophosphate	−6.50226
Ophthalmic acid	−6.28422
Deoxyguanosine	−5.83886
Azelaic acid	−5.37260
Pantothenic acid	−5.33000
Loliolide	−4.69992
Cinnamic acid	−4.63100
Pterin	−4.57262
4-Hydroxybenzoic acid	−4.45834
Salicylic acid	−4.29606
Benzoic acid	−4.23322

**Table 3 marinedrugs-21-00149-t003:** A 22 full factorial design for the optimization of *S. platensis* loaded phytosomes.

Factors (Independent Variables)	Levels
X_1_: PC amount (mg)	50 100
X_2_: CH amount (mg)	50 100
Responses (dependent variables)	Constraints
Y_1_: EE (%)	Maximize
Y_2_: PS (nm)	Maximize
Y_3_: PDI	Minimize
Y_4_: ZP (mV)	Maximize (absolute value)
Y_5_: Q6h (%)	Minimize

Abbreviations: EE%—entrapment efficiency percent, CH—cholesterol, PC—phospholipid, PS—particle size, PDI—polydispersity index, ZP—zeta potential, and Q6h—percent of drug released after 6 h.

**Table 4 marinedrugs-21-00149-t004:** Composition of full factorial design of *S. platensis* nanophytosomes and the EE%, PS, PDI, ZP and Q6h results.

Formula	PC (mg)	CH (mg)	EE (%)	PS (nm)	PDI	ZP (mV)	Q6h (%)
F1	50	0	37.81 ± 1.61	175.1 ± 0.77	0.43 ± 0.06	−13.60 ± 0.57	89.00 ± 3.50
F2	100	0	42.36 ± 6.04	206.36 ± 4.11	0.46 ± 0.06	−11.66 ± 0.57	79.00 ± 2.90
F3	50	50	51.19 ± 1.84	494.23 ± 48.05	0.69 ± 0.13	−11.13 ± 0.20	80.00 ± 4.00
F4	100	50	62.76 ± 1.75	598.40 ± 9.68	0.62 ± 0.07	−19.80 ± 0.49	74.00 ± 1.90

Abbreviations: EE%—entrapment efficiency percent, CH—cholesterol, PS—particle size, PDI—polydispersity index, PC—phospholipid, ZP—zeta potential, and Q6h—percentage of drug released after 6 h.

## Data Availability

The data presented in this study are available on request from the corresponding author.

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
