# Peer review of "Enhanced Wound Healing Potential of Spirulina platensis Nanophytosomes: Metabolomic Profiling, Molecular Networking, and Modulation of HMGB-1 in an Excisional Wound Rat Model"

_marinedrugs, 2023, doi:10.3390/md21030149_

Round 1
Reviewer 1 Report
Please, see the attached file for details.

Author Response
Rebuttal Letter
The authors appreciate the reviewer’s efforts in revising the manuscript entitled: “Enhanced wound healing potential of Spirulina platensis nanophytosomes: metabolomic profiling, molecular networking, modulation of HMGB-1 in an excisional wound rat model”. Please find below our responses to the reviewer’s comments.
Note
The highlighted parts are the parts that were added to the manuscript as per the reviewers’ comments.
Reviewer 1:
1- Page 2, lines 70-71: Please, add some examples of herbal medicines used in the treatment of excisional wounds.
As per the reviewer’s comment, a paragraph mentioning some examples of herbal medicines used in the treatment of excisional wounds has been added as follows: Medicinal plants such as Curcuma longa, Terminalia arjuna, Centella asiatica, Bidens Pilosa, Aloe barbadensis, and Rauwolfia serpentine have confirmed wound healing activity and are found to be effective in the treatment of wounds. Ref.: https://fjps.springeropen.com/articles/10.1186/s43094-021-00202-w
2- Page 4, line 157: I think that section 2.2.5 does not exist. Please, correct it.
Done
3- Page 4, line 162: I think that section 2.2.6 does not exist. Please, correct it.
Done
4- Page 21, line 660: Authors should specify the chemical composition of MEBO ointment
As per the reviewer’s comment, the chemical composition of MEBO ointment has been added as follows: MEBO® ointment’s active constituent is B-sitosterol (0.25%) and the base is composed mainly of sesame oil and beeswax.
5- Page 22, line 668: Authors should write the amount of BG11 used for the isolation of the algal strain.
The amount of BG11 has been added to section 4.2.2 as follows:
BG11 media (500 ml) containing 1.5 % agar were spread in Petri dishes and water samples 0.1 ml were inoculated in the media for the isolation of the algal strain.
6- Page 22, line 673: Authors should add the temperature used for the cultivation.
The temperature (25°C) used for the cultivation has been added to section 4.2.3.
7- Page 22, line 676-679: Please, add the amount of dried algal biomass and the filter used. Moreover, authors should describe the samples’ storage after freeze-drying process and the yield of product.
As per reviewer’s request the section 4.2.3. has been adjusted as follows:
The dried algal biomass (500 gm) was extracted by maceration using methanol (70%) in an ultrasonic water bath at 40℃. The extract was filtered on 0.45 µm membrane filter, and the filtrate was dried using a freeze-drying apparatus (Alpha-1-2-LDplus, Donau lab, Ukraine) set at atmospheric pressure at warm-up mode for 11 mins then 0.37 mbar at main drying mode for 4 hrs. The yield product; dark amorphous sticky mass was kept in dark vial at a temperature of -20°C.
8- Page 22, line 690: I suggest authors adding the antioxidant activity of Spirulina extract.
The antioxidant activity of Spirulina extract was not measured, as various research articles reported its antioxidant activity. It is reported that it activates cellular antioxidant enzymes, inhibits lipid peroxidation and DNA damage, scavenges free radicals, and increases the activity of superoxide dismutase and catalase (DOI: 10.1007/s00204-016-1744-5)
9- Page 24, line 776: Please, add the pH value of Spirulina platensis nanodispersions.
As per the reviewer’s comment, the pH measurement of Spirulina platensis nanodispersions has been added to methodology section (4.2.9) as follows the pH of the dispersion was measured using a pH meter (JENWAY model 350, JENWAY Ltd., UK) and the pH value has been added to the result section (2.9) as follows: 6.8 ± 0.02.
10- Page 24, line 788: Please, add the ratio dilution used and the solvent used
As per the reviewer’s comment, the dilution ratio and solvent were added as follows: …. in deionized water (1:100).
11- Page 24, line 789: Why did authors select this time (6h)? Please, specify it
We thank the reviewer for the valuable comment. In formulation optimization, we select one value as a comparative assessment to select the optimum formula using Design ExpertÒ. The selection of this time agrees with those made by many references e.g.:
- Hassan DH, Shohdy JN, El-Setouhy DA, El-Nabarawi M, Naguib MJ. Compritol-based nanostrucutured lipid carriers (NLCs) for augmentation of zolmitriptan bioavailability via the transdermal route: in vitro optimization, ex vivo permeation, in vivo pharmacokinetic study. Pharmaceutics. 2022 Jul 18;14(7):1484.
- Albash R, Abdelbary AA, Refai H, El-Nabarawi MA. Use of transethosomes for enhancing the transdermal delivery of olmesartan medoxomil: in vitro, ex vivo, and in vivo evaluation. International journal of nanomedicine. 2019;14:1953.
12- Page 24, line 793: Authors should specify the characteristics of cellulose membrane.
As per the reviewer’s comment the characteristics of cellulose membrane have been added as follows: cellulose membrane (12,000–14,000 molecular weight cutoff)
13- Page 24, line 795: The in vitro release studies were perfomed in PBS, why? I suggest authors doing this studies in simulated wound fluid (SWF)
The authors thank the reviewer for the valuable suggestion. In the current work the authors relied on published articles, which used PBS as the release medium for the investigation of in vitro release for the developed drug delivery system designed for wound healing, for example:
- Sanchez, David A., et al. "Amphotericin B releasing nanoparticle topical treatment of Candida spp. in the setting of a burn wound." Nanomedicine: Nanotechnology, Biology and Medicine 10.1 (2014): 269-277.
- Sanad, Rania Abdel-Basset, and Hend Mohamed Abdel-Bar. "Chitosan–hyaluronic acid composite sponge scaffold enriched with Andrographolide-loaded lipid nanoparticles for enhanced wound healing." Carbohydrate Polymers 173 (2017): 441-450.
- Adeli-Sardou, Mahboobeh, et al. "Controlled release of lawsone from polycaprolactone/gelatin electrospun nano fibers for skin tissue regeneration." International journal of biological macromolecules 124 (2019): 478-491.
14- Page 25, line 831: In order to prepare Spirulina platensis gel, authors have chosen as polymer HPMC. Why?
The Authors have chosen HPMC as it is a non-toxic, biocompatible and biodegradable polymer, which is used extensively in various applications in the fields of pharmaceuticals and food industries (Colloids Surf. A Physicochem. Eng. Asp. (2010), 10.1016/j.colsurfa.2009.09.040).
Also, HPMC has an excellent film-forming capacity (Polymer (2001), 10.1016/S0032-3861(01)00477-3).
15- Page 8, line 244: I suggest authors rewrite the sentence “the appearance of a new band is due to…”
The authors thank the reviewer for the suggestion and the following sentence has been added: The new band at 1735 cm-1 is probably due to the C=O stretching of the ester linkage of phospholipids. All the above confirm the successful formation of the complex.
16-Page 9, line 246: Figure 6: I suggest authors replace the spectra A improving line base.
The authors will consider the suggestion.
17- Page 9, line 251: Please, add the Table on the stability study showing stability of the formulations at different
Thank you for this suggestion. It would have been interesting to explore this aspect. However, in our study, stability was evaluated by comparing the initial measurements with results obtained after one year storage. Our work is similar to other published articles e.g.:
- Pooja Jain, Mohamad Taleuzzaman, Chandra Kala, Dipak Kumar Gupta, Asad Ali & Mohammed Aslam (2021) Quality by design (Qbd) assisted development of phytosomal gel of aloe vera extract for topical delivery, Journal of Liposome Research, 31:4, 381-388, DOI: 10.1080/08982104.2020.1849279.
18- Page 17, line 408: I think that section 4.3 does not exist. Please, correct it.
Done
19- Page 17, line 413: table 4 is not correct. Please, correct it.
Done

Reviewer 2 Report
The manuscript from Refai and collaborators describes the development and characterization of nanophytosomal formulation loaded with dried hydroalcoholic extract of Spirulina platensis. The authors also reported the metabolic profile of the extract and performed docking studies with the identified compounds. Finally, the healing action of formulation was evaluated using rats.
This manuscript is well written and presented. It is suitable for publication in this high-quality journal after few adjustments:
Minor issues:
- Please use italics for species names.
- Please use the abbreviated form after the first citation of species names.
- Please include the groups description at the beginning of the section 2.10, as the results section is presented before the methodology section.
- Please also include the groups descriptions at each figure legend.
- The authors should present the discussion as a single section and include the appropriate references when is suitable. The discussion is also too long.
- The section 3.1 is presented as a a single paragraph, please split it to improve the readability and understanding.
The suggestions are also provided in the attached pdf.

Author Response
Rebuttal Letter
The authors appreciate the reviewer’s efforts in revising the manuscript entitled: “Enhanced wound healing potential of Spirulina platensis nanophytosomes: metabolomic profiling, molecular networking, modulation of HMGB-1 in an excisional wound rat model”. Please find below our responses to the reviewer’s comments.
Note
The highlighted parts are the parts that were added to the manuscript as per the reviewers’ comments.
Reviewer 2:
1-The existing experimental results are difficult to prove that the optimal nanoparticles have been successfully prepared, Because the PDI values of all nanoparticles are very large, much larger than the PDI value that is considered to be good stability and dispersion, the latter generally needs to be less than or equal to 0.2
We fully agree with the valuable comment of the reviewer, however, the PDI values were considered still acceptable as they were below 0.7 according to many references, we found in the literature e.g. doi: 10.3390/pharmaceutics10020057, https://doi.org/10.1016/B978-0-12-814130-4.00011-7.
At the same time the relatively high PDI is usually observed in vesicles prepared using thin film hydration method not followed by filtration or size reduction technique such as sonication or high-shear mixing as mentioned in the following article
- Salama, A.H. and Aburahma, M.H., 2016. Ufasomes nano-vesicles-based lyophilized platforms for intranasal delivery of cinnarizine: preparation, optimization, ex-vivo histopathological safety assessment and mucosal confocal imaging. Pharmaceutical Development and Technology, 21(6), pp.706-715.
2- The authors did not explain why the addition of CH significantly increased the particle size and PDI of the nanoparticles
In the manuscript on page 20 lines 535-541 the suggested explanation for the increased particle size due to the addition of CH is already written: “The remarkable increase in PS of the formulae containing CH, namely F3 and F4, maybe also attributed to that including cholesterol into the bilayer alters the lipid vesicles’ geometrical packaging, which would accordingly affect lipid vesicle size, the curvatures and rigidity of surface bilayer[41, 42]. Furthermore, SuÅ‚kowski et al. (2005) stated that when cholesterol is incorporated in the vesicular membrane, the distance between the phospholipid chains will increase and the possibility of interaction between electronic shells of phospholipids’ polar head groups in the bilayer will become limited [43].”
The change in the geometry of the vesicles due to CH and accordingly the increased size might have affected the particles’ dispersibility to higher PDI values. This finding could be also noticed in other works e.g. (Tefas, Lucia Ruxandra, et al. "Quercetin-loaded liposomes: formulation optimization through a D-optimal experimental design." Farmacia 63.1 (2015): 26-31.)
3-The authors appear to have used the wrong method to calculate the encapsulation rate of the drug, and suggest recalculating the encapsulation rate of the drug with reference to the following literature; Small 2021, 2101495; Chemical Engineering Journal 426 (2021) 131919
We have not calculated the encapsulation rate but we calculated the entrapment efficiency % (EE%) using the equation published in many works. In literature, the EE% could be calculated directly by determining the amount of entrapped drug in the nanocarriers divided by the total amount of the drug e.g.
https://doi.org/10.1016/j.biomaterials.2006.03.006,https://doi.org/10.3109/10837450.2015.1048553)
or indirectly by subtracting the amount of the drug in the supernatant (the free encapsulated drug) from the total amount of drug and then dividing the result by the total e.g.
(https://doi.org/10.1016/j.cbi.2017.04.026, https://doi.org/10.1016/j.ijpharm.2013.03.021,
4-The authors did not provide any data to prove that SPNP hydrogels had been successfully prepared
Thank you for pointing this out. We have included the characterization of the gel in the supplementary file and in the manuscript as follows:
2.10. Characterization of the SPNP-gel
The optimized formula was incorporated in a 2% HPMC gel (SPNP-gel). The gel revealed optimum results for topical application to open wounds with respect to organoleptic properties, pH, drug content, rheological behavior and spreadability. The detailed characterization of the gel is provided in the supplementary file.
The numbering of the following sections was adjusted accordingly.
Also, in the methodology section 4.2.17. the following has been added: The detailed characterization of the gel is provided in the supplementary file.
5- The authors did not explain why only the drug release behavior of the nanoparticles after 6 hours was examined
We thank the reviewer for the valuable comment. In formulation optimization, we select one value as a comparative assessment to select the optimum formula using Design ExpertÒ. The selection of this time agrees with those made by many references e.g.:
- Hassan DH, Shohdy JN, El-Setouhy DA, El-Nabarawi M, Naguib MJ. Compritol-based nanostrucutured lipid carriers (NLCs) for augmentation of zolmitriptan bioavailability via the transdermal route: in vitro optimization, ex vivo permeation, in vivo pharmacokinetic study. Pharmaceutics. 2022 Jul 18;14(7):1484.
- Albash R, Abdelbary AA, Refai H, El-Nabarawi MA. Use of transethosomes for enhancing the transdermal delivery of olmesartan medoxomil: in vitro, ex vivo, and in vivo evaluation. International journal of nanomedicine. 2019;14:1953.
The graph representing the release profiles of the studied formulations has been added to the supplementary file.
6- The biocompatibility of the prepared nanoparticles and hydrogels also needs to be further evaluated.
Regarding the biocompatibility, the histopathological evaluation of the skin tissues treated with nanoparticles and hydrogel showed no signs of increased irritation in comparison to the untreated group; on the contrary, there was improved healing and downregulated inflammation, suggesting the biocompatibility of the used formula as a topical application. The authors thank the reviewer for making this point, and we will consider it in our future research.

Reviewer 3 Report
This work reports on the preparation of Spirulina platensis nanophytosomes, the SPNP gel, and assesses its potential in promoting excisional wound healing. The authors further examined the mechanisms by which the SPNP gel promotes wound healing. The main issues of this work are as follows. 1) The existing experimental results are difficult to prove that the optimal nanoparticles have been successfully prepared, Because the PDI values of all nanoparticles are very large, much larger than the PDI value that is considered to be good stability and dispersion, the latter generally needs to be less than or equal to 0.2; ​2) The authors did not explain why the addition of CH significantly increased the particle size and PDI of the nanoparticles;3) The authors appear to have used the wrong method to calculate the encapsulation rate of the drug, and suggest recalculating the encapsulation rate of the drug with reference to the following literature; Small 2021, 2101495; Chemical Engineering Journal 426 (2021) 131919; 4) The authors did not provide any data to prove that SPNP hydrogels had been successfully prepared; 5) The authors did not explain why only the drug release behavior of the nanoparticles after 6 hours was examined; 6)The biocompatibility of the prepared nanoparticles and hydrogels also needs to be further evaluated;
Author Response
Rebuttal Letter
The authors appreciate the reviewer’s efforts in revising the manuscript entitled: “Enhanced wound healing potential of Spirulina platensis nanophytosomes: metabolomic profiling, molecular networking, modulation of HMGB-1 in an excisional wound rat model”. Please find below our responses to the reviewer’s comments.
Note
The highlighted parts are the parts that were added to the manuscript as per the reviewers’ comments.
Reviewer 3:
1- Please use italics for species names.
Done
2- Please use the abbreviated form after the first citation of species names.
Done
3-Please include the groups description at the beginning of the section 2.10, as the results section is presented before the methodology section.
Done as follows:
2.10. Effect of SPNP-gel topical application on wound contraction rate
To assess the healing potential of Nano-Spirulina (SPNP)-gel after wound induction, contraction rate was observed and documented by capturing images at different time intervals (0, 3, 6, 9, 12, and 14 day). The analysis revealed that MEBO® ointment (GpIII/standard group; injured rats in this group were treated topically with MEBO® 0.2 g two times/day), Spirulina gel(GpIV; rats in this group were treated topically with Spirulina platensis gel as the standard group ), and SPNP gel groups(GpV; rats in this group were treated topically with SPNP-gel as the standard group) markedly increased contraction rate as compared to the +ve control group (wounded un treated rats) starting from day 3 after surgery. Besides, the wound-contraction rate of SPNP-gel was significantly higher than that of other treatment groups at all the time intervals evaluated.
4-Please also include the groups descriptions at each figure legend.
However, we believe the ligands will be quite verbose, thus in response to the reviewer's comment, the following paragraph has been added to all the fig. ligands:
GpI/control negative (-ve): non-wounded rats, GpII/control positive (+ve): untreated wounded rats, GpIII/standard group: wounded rats treated with MEBO®, GpIV/ S. platensis gel: wounded rats treated with S. platensis gel, and GpV/ Nano-Spirulina (SPNP)-gel: wounded rats treated with SPNP-gel
5- The authors should present the discussion as a single section and include the appropriate references when is suitable. The discussion is also too long.
The discussion section's titles were removed so that it could be presented as a single section, but this study uses a variety of methods and assays that must all be discussed, supported, and evaluated in comparison to earlier reports. Each section of the discussion is crucial to the study's validity and reproducibility. Thus, we are apologetic for the long discussion.
6- The section 3.1 is presented as a a single paragraph, please split it to improve the readability and understanding.
Done

Round 2
Reviewer 1 Report
Minor comments:
page 2, lines 72-73: Authors should correctly write the name of plants e.g. Curcuma Longa L.
page 23, line 722: Authors should write the yield of product obtained after filtration
Author Response
Response Letter (round two)
The authors appreciate the reviewer’s efforts for revising the manuscript entitled: “Enhanced wound healing potential of Spirulina platensis nanophytosomes: metabolomic profiling, molecular networking, modulation of HMGB-1 in an excisional wound rat model”. Please find below our responses to the reviewer’s comments.
Note
The newly highlighted parts in blue are the parts that were added to the manuscript as per the reviewers’ comments.
Reviewer 1:
-page 2, lines 72-73: Authors should correctly write the name of plants e.g. Curcuma Longa L.
Done as follow “Providing an efficient agent for wound dressing is always an urgent concern in modern medicine. Herbal medicines and phytoconstituents have been demonstrated to be useful in the treatment of a wide range of health conditions including wound healing. Medicinal plants such as Curcuma longa L., Terminalia arjuna Roxb., Centella asiatica L., Bidens pilosa L., Aloe barbadensis Mill., and Rauwolfia serpentine L. have confirmed wound healing activity and are found to be effective in the treatment of wounds”
-page 23, line 722: Authors should write the yield of product obtained after filtration
Done as follow “The yield product (7.7 gm)”
